# Efficient Discriminative Joint Encoders for Large Scale Vision-Language Re-ranking

**Mitchell Keren Taraday**[1]* **Shahaf Wagner**[1]* **Chaim Baskin**[1]
[1]INSIGHT Lab, Ben-Gurion University of the Negev, Israel

## Abstract

Multimodal retrieval still leans on embedding-based models like CLIP for fast vector search over pre-computed image embeddings. Yet, unlike text retrieval where joint-encoder re-rankers are standard, comparable vision–language re-rankers are largely absent. We find that seminal joint encoders such as BLIP are severely bottlenecked by an expensive visual feature-extraction stage, preventing practical deployment at scale. Motivated by this bottleneck, we introduce *EDJE*, an **E**fficient **D**iscriminative **J**oint **E**ncoder that precomputes vision tokens offline and compresses them via a lightweight attention-based adapter, so online inference runs only a compact joint encoder over a small set of visual tokens plus the text. *EDJE* preserves strong retrieval performance while drastically reducing storage and online compute, enabling high-throughput inference. Specifically, *EDJE* processes 50k image–text pairs/second while requiring 49kB of disk storage per image, matching prior art on Flickr (zero-shot) and COCO (fine-tuned) retrieval. [1].

## 1 Introduction

Large-scale multimodal retrieval — finding the most relevant images for a text query, or retrieving descriptive text given an image — is a fundamental challenge in vision–language modeling. Its importance spans a wide range of applications, including web-scale image search, multimodal dataset curation, content moderation, and retrieval-augmented generation. Because such applications often involve searching across millions of candidates, retrieval systems must be *both* efficient and accurate.

A major breakthrough came with the emergence of models that align visual and textual modalities within a shared embedding space, such as CLIP (Radford et al., 2021). By enabling efficient similarity search through simple vector comparisons, these models made content-based large-scale retrieval feasible. Beyond retrieval, they have also shown strong generalization to tasks such as zero-shot

---

*Equal contribution.

[1]The paper's website, including code, is publicly available at https://shahafwa.github.io/EDJE/

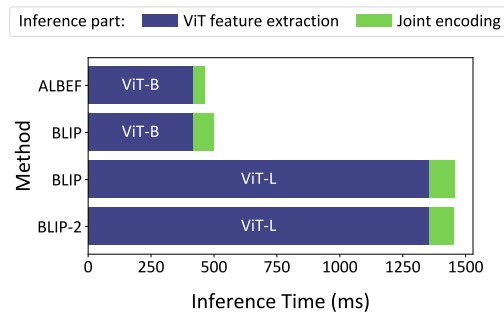
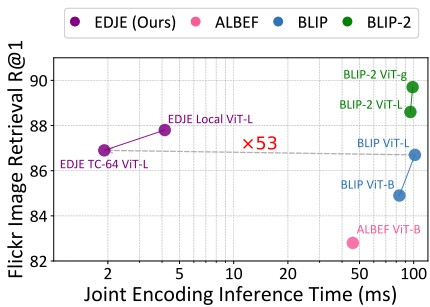

| (a) Inference time breakdown | (b) Retrieval vs. encoding time |

Figure 1: **Inference efficiency and retrieval performance.** (a) Methods with strong discriminative capabilities are dominated by costly ViT feature extraction, prohibiting their practical use for re-ranking. (b) *EDJE* achieves competitive zero-shot retrieval performance with up to $53\times$ faster inference. Its token compression makes storing visual features practical, enabling large-scale retrieval.

image classification, inspiring rapid improvements of this paradigm (Wang et al., 2022; Zhai et al., 2023; Cherti et al., 2023; Fang et al., 2023a;b; Tschannen et al., 2025).

In parallel, the remarkable success of large language models (LLMs) has driven efforts to integrate vision, enabling instruction-following over multimodal inputs. Early approaches (Chen et al., 2020; Li et al., 2021; 2022; 2023) aimed to build foundation vision–language models (VLMs) capable of both generative and discriminative tasks. However, the research community has shifted its interest towards generative-only models, typically by coupling a vision encoder with a pre-trained LLM (Liu et al., 2023; 2024; Wang et al., 2024; Gemma, 2025). This shift has effectively divided the research community into two main directions: (1) advancing embedding-based models for vision–language alignment, and (2) improving text generation over multimodal inputs — leaving the discriminative potential of joint encoders largely underexplored.

Unlike embedding-based models, joint encoders process both modalities together, allowing *richer cross-modal interactions*. Prior work (Li et al., 2021; 2022; 2023) has shown that such models can significantly improve cross-modal retrieval performance by re-ranking the top-$k$ candidates retrieved by an embedding model. However, their adoption in practical retrieval pipelines has remained limited; each candidate pair must be evaluated independently, and existing architectures are slow. In particular, these models rely on heavy, high-resolution CLIP-style vision backbones to extract highly expressive image features that poses a severe efficiency bottleneck (Figure 1a). This raises a central question:

*Can we harness the benefits of joint modeling while achieving the efficiency required for large-scale retrieval?*

To this end, we introduce *EDJE*, an efficient discriminative joint encoder that allows fine-grained cross-modal interactions without requiring online visual feature extraction. The core idea is to shift visual feature extraction offline: images are encoded once and stored on disk; at query time a compact encoder-only language model jointly processes these with text tokens to produce a re-ranking score. We further improve scalability by introducing a lightweight *token-compression adapter* that reduces the number of cached vision tokens. Instead of storing the full sequence produced by the vision backbone, the adapter utilizes a small set of learnable queries that aggregates the most relevant information through cross-attention and projects them to the embedding space of the joint encoder. This compressed representation substantially lowers storage requirements and decreases the number of tokens the joint encoder must process at query time.

Empirically, *EDJE* consistently improves zero-shot retrieval when paired with a variety of embedding-based models, spanning multiple visual backbones and input resolutions. This demonstrates its modularity as a drop-in re-ranker that can enhance retrieval pipelines regardless of the underlying embedding model. Moreover, when equipped with a strong visual backbone such as SigLIP2 (Tschannen et al., 2025), *EDJE* surpasses or matches the retrieval performance of prior joint encoders on standard benchmarks (Flickr30k zero-shot; MS-COCO fine-tuned) (Plummer et al., 2015; Lin et al., 2014), while operating with substantially greater efficiency (Figure 1b). Finally, we evaluate the robustness of *EDJE* under compression, quantifying the trade-off between retrieval performance and storage cost as the number of compressed tokens is reduced, and conducting further ablations on re-ranking pool size and training objectives.

**Contributions.** In this work, we address the challenge of bringing the benefits of joint vision–language modeling to large-scale retrieval while maintaining efficiency. Our main contributions are as follows:

1. We introduce *EDJE*, an efficient discriminative joint encoder that performs fine-grained cross-modal re-ranking while shifting heavy vision precomputation offline. We further propose a lightweight *token-compression adapter* that condenses vision features into a compact representation, substantially reducing storage and computation.

2. Empirically, *EDJE* demonstrates consistent gains over a variety of embedding-based models. With a strong visual backbone, *EDJE* achieves performance competitive with state-of-the-art joint encoders on standard benchmarks while operating with substantially greater efficiency.

3. We conduct comprehensive analyses of scalability and robustness, quantifying trade-offs between retrieval performance, storage costs, re-ranking pool size, and training objectives.

## 2 RELATED WORK

The success of CLIP (Radford et al., 2021) and ALIGN (Jia et al., 2021) in aligning vision and language modalities within a shared embedding space marked a breakthrough in vision–language modeling. By scaling contrastive learning to large architectures and massive image–caption datasets, these models enabled efficient vector similarity search and inspired abundance of follow-up work.

Subsequent research has been directed towards reproducing and extending this paradigm in several directions. For example, LAION-400M (Schuhmann et al., 2021) released an open dataset of paired image–caption training data. Other efforts scaled model size and data (Xu et al., 2024; Fang et al., 2023b), filtered noisy captions (Fang et al., 2023a; Gadre et al., 2023), or generated synthetic ones (Li et al., 2022; Nguyen et al., 2023; kokitsi Maninis et al., 2025). Additional work explored alternative loss functions (Zhai et al., 2023) or auxiliary objectives to enrich localization (Naeem et al., 2024; kokitsi Maninis et al., 2025) and language generation capabilities (Wan et al., 2024; Tschannen et al., 2025). Despite their efficiency and scalability, embedding-based approaches compress modalities independently (*late interaction*), limiting fine-grained cross-modal interactions.

Parallel to contrastive approaches, researchers have pursued models that process modalities *jointly*. Early systems such as LXMERT and UNITER (Tan and Bansal, 2019; Chen et al., 2020) relied on region features from R-CNN (Ren et al., 2015) combined with text embeddings. LightningDOT (Sun et al., 2021) relies on these methods to perform re-ranking with pre-computed region-level representations to enable feasible storage. However, because each region is collapsed into a single vector, such re-ranker behaves much closer to an embedding model rather than a true joint encoder that sees the full image. In practice, this leads to performance that now lags behind modern embedding models such as SigLIP2 (Tschannen et al., 2025).

The emergence of vision transformers made combining vision and language modalities more straightforward as both modalities are represented as a sequence of tokens. Consequently, some works aimed at creating transformer models capable of processing both images and texts jointly (Wang et al., 2021; 2022). Such models require heavy pre-training by masking either text or both text and image tokens. Another line of work introduced cross-attention architectures such as ALBEF, BLIP, BLIP-2, and CoCa (Li et al., 2021; 2022; 2023; Yu et al., 2022), which fuse pretrained vision encoders and language models through cross-attention layers. These joint encoders not only unify discriminative and generative modeling, but also consistently outperform embedding-based models on discriminative tasks. In particular, multimodal retrieval performance can be significantly enhanced by re-ranking embedding-based results with a joint encoder (Li et al., 2021; 2022; 2023), echoing common practices in text retrieval where cross-encoders are widely used (Chen et al., 2024; Zhang et al., 2024).

A more recent trend is the integration of pretrained vision encoders into large language models (LLMs), yielding generative vision–language models (VLMs). Methods such as LLaVA (Liu et al., 2023) introduce a lightweight projection that maps vision tokens into the LLM embedding space,

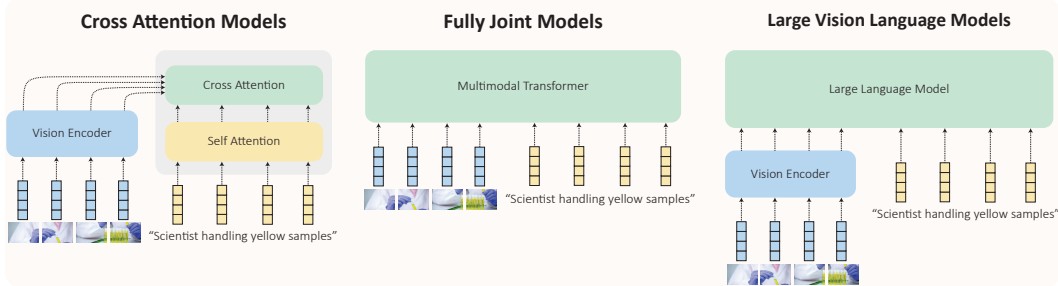

Figure 2: Taxonomy of vision–language joint encoders. Left: Cross-attention models integrate modalities through cross-attention layers interleaved with textual self-attention (Li et al., 2021; 2022; 2023). Middle: Joint foundation models such as BEiT-3 (Wang et al., 2022) employ unified self-attention over native visual and textual tokens, enabling full cross-modal interaction. Right: Modern generative VLMs (Liu et al., 2023) combine a pretrained vision encoder with a large language model, tuning the latter to process projected vision tokens as if they originated from text.

followed by fine-tuning on curated captioning datasets. Variants extend this with parameter-efficient fine-tuning (Hu et al., 2022) and supervised fine-tuning (Liu et al., 2024; Abdin et al., 2024; Zohar et al., 2025; Microsoft et al., 2025). While this approach makes it simpler to integrate various modalities into highly optimized LLMs (Zohar et al., 2025; Microsoft et al., 2025), it often emphasizes instruction-following and generation at the expense of discriminative power.

A taxonomy of contemporary vision-language joint encoders is provided in Figure 2.

## 3  TOWARDS EFFICIENT JOINT ENCODERS

We now build up the design of our efficient discriminative joint encoder (*EDJE*) step by step. First, we examine why existing multimodal joint encoders remain impractical for retrieval, pinpointing the vision backbone as the critical bottleneck (Section 3.1). Next, we show how precomputing vision features offers an appealing solution, while also introducing a new challenge: the considerable cost of storing all tokens (Section 3.2). Next, we discuss an efficient integration of vision and language modalities through a compact joint encoder (Section 3.3). Finally, we present a token-compression adapter that resolves the storage challenge by compressing long sequences of vision tokens into a compact set of expressive tokens (Section 3.4).

### 3.1  ON THE ABSENCE OF MULTIMODAL RE-RANKERS

Existing joint encoders such as BLIP and BLIP-2 (Li et al., 2022; 2023) achieve strong performance but rely on visual feature extraction through large backbones like ViT-B/16 (384) and ViT-L/16 (384). This reliance introduces a severe bottleneck: encoding a batch of 64 images requires about 400 ms with ViT-B and nearly 1,400 ms with ViT-L on an A6000 GPU - before any cross-modal interaction even occurs. Specifically, for the BLIP family, the visual feature extraction alone consumes 83% of inference time in the ViT-B case and 93% with ViT-L. In practice, such inference times make it infeasible to use these models for retrieval, where thousands of candidates must be re-ranked per query. In comparison, the most downloaded text re-ranker model in HuggingFace[2] is based on the MiniLM architecture (Wang et al., 2020), has just 22M parameters and processes a similar batch of full-context sequences in under 60 ms, an order of magnitude faster. This gap explains the absence of multimodal re-rankers in real-world systems: the cost of extracting visual features alone is prohibitive.

### 3.2  PARADIGM SHIFT: VISION PRECOMPUTATION

With the vision backbone identified as the bottleneck, we next ask: must vision features always be extracted online? Cross-attention-based models and VLMs suggest otherwise: since the vision encoder operates purely on images, its output can be cached and reused. Thus, we propose treating the vision encoder as a preprocessing stage, with vision tokens computed and stored to disk *offline*.

For a standard ViT-B (Dosovitskiy et al., 2021) projecting each $16 \times 16$ patch to a $d = 384$ embedding stored in FP16 occupies the same space as the uncompressed 8-bit RGB image[3]. These token representations can reside on disk rather than in memory, as in late-interaction models like ColBERT (Khattab and Zaharia, 2020) and ColPali (Faysse et al., 2025). Under fixed token dimensionality, scaling the vision encoder improves representation quality while leaving per-image storage unchanged—shifting heavy computation offline without increasing online cost. However, storing *all* tokens is intractable at scale: raw image size is typically too large, amounting to terabytes for web-scale databases. This problem motivates the development of strategies to compress the visual features.

> **Key takeaway:** Precomputing vision tokens moves expensive computation offline, enabling powerful vision encoders without slowing inference. However, it comes at large storage costs, motivating methods to compress the visual features.

---

[2] https://huggingface.co/cross-encoder/ms-marco-MiniLM-L6-v2
[3] $16^2 \times 3 \times 8$ bits per patch vs. $384 \times 16$ bits per token.

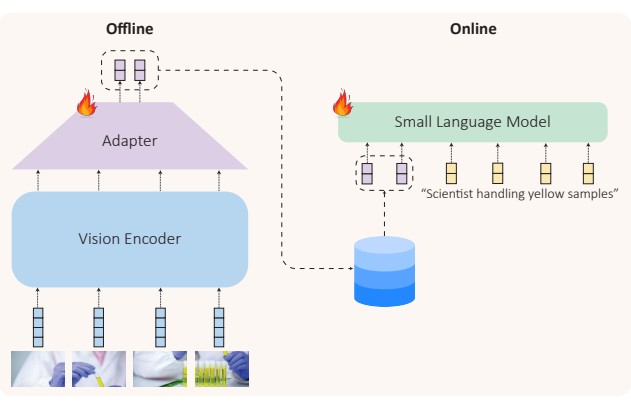 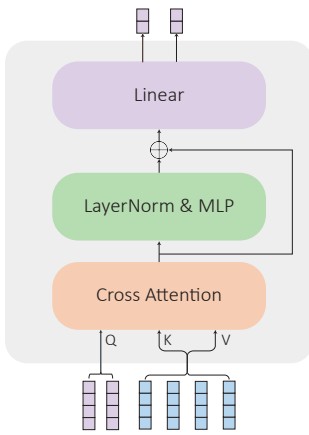

(a) A high-level view of *EDJE*                    (b) Token-compression adapter

Figure 3: *EDJE* architecture overview and adapter. (a) Offline stage (left): images are encoded by the vision encoder and projected by the adapter into a compact set of tokens compatible with the language model. Online stage (right): the small language model consumes the compressed tokens together with text. (b) Token-compression adapter: cross-attention utilizes $k$ universal query tokens that act as feature extractors acting on the visual tokens. The MLP projects the extracted features to the embedding space of the language model.

## 3.3 INTEGRATING THE VISION MODALITY

Once visual tokens are computed, the question becomes how best to integrate them with text. Considering how the vision tokens are integrated into "cross-attention" models versus how they are integrated into large vision-language-models, we make an interesting observation: while in "cross attention" models vision tokens are considered in the cross attention layers, large VLMs instead project vision tokens into the language embedding space and *concatenate* them with text tokens; this allows the cross-modal interaction to be handled entirely by self-attention layers. In our setting, the large language model can be replaced with a compact, efficient language model to meet throughput targets. This yields an architecture with many benefits: (i) Fast: the language model can be as small as MiniLM (Wang et al., 2020) or any other efficient language model. (ii) Modular: any ViT-based vision encoder can be paired with any pre-trained language model via a lightweight adapter as a bridge between modalities. (iii) Expressive: modern vision encoders produce highly expressive tokens that capture both semantics and local spatial structure (Tschannen et al., 2025). (iv) Data efficient: only the adapter has to be trained from scratch. In the VLM literature it has been observed that the language model and vision encoder require minimal tuning (Liu et al., 2023; 2024; Abdin et al., 2024). A high-level view of *EDJE* is given in Figure 3a.

> **Key takeaway:** Replacing the LLM in a typical VLM with a small, efficient language model yields a joint-encoder architecture well suited for discriminative modeling: fast, expressive, modular and data efficient.

## 3.4 VISION-LANGUAGE ADAPTER LAYER

The adapter projects cached vision tokens into the language embedding space. It has been demonstrated in the VLM literature that even very simple adapters — linear layers (Liu et al., 2023) or multi-layer perceptrons (MLPs) (Liu et al., 2024) are surprisingly effective despite their parameter count. However, local adapters map each vision token to one language token, limiting flexibility: vision encoders with larger context improve expressiveness but proportionally inflate storage.

To address this, we propose an attention-based token compression layer that compresses vision token sequences into a compact set of tokens. Specifically, we introduce a set of $m$ learnable universal

query tokens $Q = [q_1, \ldots, q_m]$ that attend over the $n$ vision encoder tokens $X = [x_1, \ldots, x_n]$:

$$K = XW_K, \quad V = XW_V \tag{1}$$
$$H = \text{MultiHeadAttention}(Q, K, V) \tag{2}$$

The output $H = [h_1, \ldots, h_m]$ is composed of $m$ tokens that share their dimensionality with the vision encoder, $h_i \in \mathbb{R}^{d_{\text{vision}}}$. It is useful to regard the query tokens as a universal feature extractors, that softly select visual features most relevant for semantic matching. These states are than passed through a standard residual block and projected into the language model embedding space $\mathbb{R}^{d_{\text{language}}}$ through a simple linear projection:

$$O = H + \text{MLP}(\text{LayerNorm}(H)) \tag{3}$$
$$Y = OW_{\text{proj}} \qquad W_{\text{proj}} \in \mathbb{R}^{d_{\text{vision}} \times d_{\text{language}}} \tag{4}$$

This mechanism provides a more flexible way of integrating visual information. Note that it generalizes attention pooling strategies used in embedding models (Zhai et al., 2023) and has some connection to the Q-Former layer (Li et al., 2023). The token compression layer is depicted in Figure 3b.

We refer to *EDJE* with a simple MLP adapter layer as the "local" variant vs. when equipped with a token-compression adapter which we refer to as the "token-compressed" variant.

> **Key takeaway:** The proposed token compression layer substantially decreases storage costs, seamlessly enabling vision encoders with longer context, higher input resolution and capacity.

## 4 EFFECTIVE DISCRIMINATIVE TRAINING

To obtain a joint encoder with strong discriminative performance, a natural choice is to optimize it for image–text matching. This involves determining the correspondence $f_\theta(t, v)$ of a given image $v$ to a textual description $t$, where each positive pair must be contrasted against non-matching samples. Directly training the encoder in this way poses several challenges:

**Negative pairs.** While obtaining positive pairs from paired image–caption datasets is straightforward, selecting negative pairs is considerably more challenging. Random negatives are typically too easy, failing to distinguish fine-grained matches from loosely related examples. Conversely, mining negatives with another model introduces an inevitable dependence on that model's quality. To address this, we adopt an in-batch hard-negative mining strategy utilizing an embedding model (matching to the vision encoder). For each mini-batch $\mathcal{B}$, we compute pairwise similarities between all texts and images using the embedding model, obtaining weak similarity matrix $\tilde{S}_{ij}$. For every anchor pair, we then select the top-$k$ most similar (non-anchor) images and texts according to $\tilde{S}_{ij}$ as negatives. This approach effectively exposes the joint encoder to the most confusable candidates without requiring full pairwise late interaction. Although this procedure may introduce occasional false negatives, in practice the abundance of informative negatives improves discriminative performance.

**Vision-language alignment.** While image–text matching is the central task of interest, it only provides a limited signal for aligning vision–language features and learning a meaningful global representation. Large vision–language models like LLaVA (Liu et al., 2023; 2024) achieve this alignment through a pre-training phase in which the model is encouraged to reproduce the caption matching a certain image. Inspired by the pre-training phase of such models and to exploit the bidirectionality of our joint encoder, we employ masked language modeling with aggressive text-only masking. To strengthen the dependence of the [CLS] token on textual inputs, we introduce a projection layer on top of the [CLS] representation and encourage it to recover the text embedding of the underlying embedding model when provided with text-only inputs.

Thus, our pre-training strategy jointly optimizes three heads on top of a shared backbone:

1. **Image–text matching:** binary classification over matched image–caption pairs vs. mined in-batch hard negatives.

2. **Masked language modeling:** we mask $50\%$ of caption tokens and predict the masked tokens given visual tokens and unmasked text.

3. **Text-embedding recovery:** we opt for recovering the embeddings of the text encoder $g$ paired with the vision encoder using a cosine objective $\mathcal{L}_{\text{text}}(\theta) = 1 - \cos\big(\mathbf{f}_\theta(t),\, \mathbf{g}(t)\big)$

**Local-to-compressed distillation.** To further enhance the performance of the token-compressed models we perform logit-level knowledge distillation using the local-adapter model variant as a teacher. Specifically, we encourage the token-compressed model (student) to imitate the image-text matching logits of the full-adapter joint encoder (teacher). For each positive, negative-image and negative-text pair we consider the binary cross-entropy between student and teacher predictions:

$$\mathcal{L}_{distil} = -\big[y_t \cdot \log(\hat{y}_s) + (1 - y_t) \cdot \log(1 - \hat{y}_s)\big]$$

with $y_t = \sigma\big(s_{teacher}(t, v)\big)$ and $\hat{y}_s = \sigma\big(s_{student}(t, v)\big)$ where $\sigma$ is the sigmoidal function and $s(t, v)$ denotes the similarity logit that corresponds to $t$ and $v$.

## 5 EXPERIMENTS

To goal of this section is to extensively investigate the empirical benefits of integrating *EDJE* into large scale retrieval pipelines. Specifically, we aim to address the following questions:

**(Q1)** Can *EDJE*, as a minimal-scale joint encoder, beat highly-optimized embedding models?

**(Q2)** How *EDJE* compares with existing joint encoders in terms of performance and efficiency?

**(Q3)** What is the significance of each component constituting *EDJE*?

### 5.1 EXPERIMENTAL SETUP

We train *EDJE* using a two-phase protocol consisting of pre-training and fine-tuning phases as described in Section 4. During both phases we freeze the vision encoder and train only the adapter of interest and the language model to process both modalities. We experiment with a variety of vision encoder families at multiple scales and input resolutions, including CLIP (Radford et al., 2021), DFN (Fang et al., 2023a), MetaCLIP (Xu et al., 2024) and SigLIP2 (Tschannen et al., 2025). Except for SigLIP2, we use the penultimate-layer hidden states as the vision-encoder output. The language model is fixed to be MiniLM-L12-uncased (Wang et al., 2020) in all experiments. To ensure fair comparison, we use the smaller dataset mixture of BLIP for training; the pre-training data is composed of CC12M (Changpinyo et al., 2021), CC3M (Sharma et al., 2018), SBU (Ordonez et al., 2011), Visual Genome (Krishna et al., 2016), and COCO (Lin et al., 2014), totaling 14M image–caption pairs while fine-tuning only utilizes COCO. Full training hyperparameters are provided in Appendix A.

For evaluation, we follow a two-stage retrieval pipeline: for each query, we first retrieve the top-$k$ candidates using embedding-based retrieval with a CLIP-like model. These candidates are then re-ranked by *EDJE*, which jointly processes image token embeddings and captions. Unless otherwise stated, we fix the pool-size to $k = 10$. We report both text-to-image (T2I) and image-to-text (I2T) performance under Recall@$\{1, 5, 10\}$, consistent with prior foundation model benchmarks (Li et al., 2021; 2022; 2023; Wang et al., 2022). Since most embedding-based models report other or non-retrieval metrics, we reproduce them using the OpenCLIP framework (Ilharco et al., 2021), verifying agreement with their reported numbers before presenting the aforementioned metrics. We evaluate on Flickr30k (Plummer et al., 2015) for zero-shot retrieval and on COCO (Lin et al., 2014) for fine-tuned retrieval. Following standard practice, we adopt the Karpathy split for COCO and the standard test split of 1,000 images for Flickr30k, each paired with five captions. We additionally evaluate *EDJE* under a more challenging retrieval setup in which all training images and captions are inserted into the candidate pool, following LightningDOT (Sun et al., 2021). This makes the task considerably more challenging and resemble real-world scenarios. We provide the full details and results in Appendix F.

Table 1: **Zero-shot retrieval results on Flickr30K.** We report Recall@1/5/10 for text-to-image and image-to-text tasks across four backbones (CLIP, DFN, MetaCLIP and SigLIP 2) using various ViT scales and resolutions. Rows marked red represent *EDJE* with the corresponding ViT backbone.

| Model | ViT variant | Res. | Text-To-Image | | | Image-To-Text | | |
|---|---|---|---|---|---|---|---|---|
| | | | R@1 | R@5 | R@10 | R@1 | R@5 | R@10 |
| **CLIP** | ViT-B/16 | $224^2$ | 62.1 | 85.6 | 91.8 | 81.3 | 96.1 | 98.3 |
| | | | 76.8 | 90.7 | 91.7 | 91.1 | 98.2 | 98.4 |
| | ViT-L/14 | $224^2$ | 65.2 | 87 | 92.1 | 85.1 | 97.1 | 98.9 |
| | | | 80.6 | 91.4 | 92.2 | 92.8 | 98.5 | 98.9 |
| | ViT-L/14 | $336^2$ | 67.7 | 88.8 | 93.3 | 86.7 | 98.2 | 99 |
| | | | 81.9 | 92.8 | 93.3 | 93.8 | 98.6 | 99.9 |
| **DFN** | ViT-L/14 | $224^2$ | 75.1 | 92.7 | 96 | 90 | 98.6 | 99.4 |
| | | | 77.5 | 94 | 96.1 | 91.1 | 98.4 | 99.4 |
| **MetaCLIP** | ViT-L/14 | $224^2$ | 76.3 | 93.6 | 96.3 | 90.6 | 98.5 | 99.5 |
| | | | 79.2 | 94.5 | 96.3 | 91.9 | 99.0 | 99.5 |
| **SigLIP 2** | ViT-B/16 | $384^2$ | 82.1 | 95.5 | 97.9 | 93.8 | 99.3 | 99.9 |
| | | | 84.3 | 96.6 | 97.9 | 94.3 | 99.9 | 99.9 |
| | ViT-L/16 | $384^2$ | 82.3 | 96 | 98 | 94.8 | 99.6 | 99.9 |
| | | | 87.8 | 97.3 | 98 | 96.5 | 99.8 | 99.9 |

## 5.2 MAIN RESULTS

We first examine whether *EDJE*, when considered as a lightweight joint encoder with minimal capacity, can substantially improve retrieval performance over embedding-based pipelines. To this end, we deploy the local variant as a top-$k$ re-ranker: for each embedding model tested, *EDJE* reuses its vision backbone and pairs it with MiniLM as the shared language encoder. We evaluate zero-shot retrieval performance on Flickr30k with standard text-to-image and image-to-text retrieval tasks. The results are summarized in Table 1.

*EDJE* boosts retrieval performance across all tested embedding models, emphasizing the potential of integrating re-rankers to existing retrieval pipelines. Specifically, we observe massive gains for the original CLIP (Radford et al., 2021) model, with Recall@1 improvements of up to 15% for image retrieval and 10% for text retrieval. Noticeable gains are also obtained for the SigLIP2 backbone, despite it being a highly optimized state-of-the-art embedding model. The improvements for DFN and MetaCLIP are less noticeable; however, DFN relies on a filtering network fine-tuned on Flickr.

We next assess *EDJE* when considered as a practical alternative to prior joint encoders (Li et al., 2021; 2022; 2023), fixing the visual backbone to SigLIP 2 with a resolution of $384^2$ to match their setup. To ensure fairness, we evaluate both local and token-compressed variants under a cached-vision regime. Namely, we assume that the visual features are precomputed, so that only the online joint-encoding part is considered allowing us to compare against previous methods. Under this setup we compare methods along several axes: retrieval accuracy (Recall@1 on Flickr, zero-shot, and COCO, fine-tuned, for both image-to-text and text-to-image), per-image storage (kilobytes), joint-encoder parameter count, online inference time (milliseconds for a batch of 64 on an A6000 GPU), and the amount of training data used. The results are summarized in Table 2.

*EDJE* achieves a favorable accuracy–efficiency profile relative to existing joint encoders. The local variant matches prior work on Flickr (zero-shot) and remains competitive on COCO (fine-tuned), while using a much smaller joint-encoder (tens of millions of parameters rather than hundreds) and substantially lower online latency. Crucially, these gains come with affordable storage costs: even in its uncompressed form it may be suitable for some use-cases, and the token-compressed variant

Table 2: **Comparison to prior art.** We compare *EDJE* in its Local and token-compressed (Compressed) variants (highlighted in red) against ALBEF, BLIP, and BLIP-2 Li et al. (2021; 2022; 2023) in both base and large configurations. The table reports retrieval performance: text-to-image and image-to-text R@1 on Flickr (zero-shot) and COCO (fine-tuned). We also report the amount of training data used. Finally, we include per-image storage, joint-encoder parameter count, and inference time for a batch of 64 samples.

| Method | Training data | Flickr-ZS T2I | Flickr-ZS I2T | COCO-FT T2I | COCO-FT I2T | Storage per image | Joint encoding parameters | Inference time (ms) |
|---|---|---|---|---|---|---|---|---|
| ALBEF [ViT-B/16] | 12M | 82.8 | 94.1 | 60.7 | 77.6 | 1,769 kB | 147M | 45.92 |
| BLIP [ViT-B/16] | 12M | 84.9 | 94.8 | 63.1 | 80.6 | 1,769 kB | 139M | 83.27 |
| BLIP [ViT-L/16] | 129M | 86.7 | 96.7 | 65.1 | 82.4 | 2,359 kB | 139M | 101.61 |
| BLIP-2 [ViT-L/16] | 400M | 88.6 | 96.9 | 66.3 | 83.5 | 2,359 kB | 167M | 98.64 |
| Local [ViT-B/16] | 12M | 84.3 | 94.3 | 60.9 | 76.1 | 442kB | 33M | 2.86 |
| Local [ViT-L/16] | 12M | 87.8 | 96.5 | 64.9 | 81 | 442kB | 33M | 4.14 |
| Compressed-128 [ViT-L/16] | 12M | 87.1 | 96.3 | 64.6 | 81 | 98kB | 33M | 2.04 |
| Compressed-64 [ViT-L/16] | 12M | 86.9 | 96.4 | 64.6 | 80.9 | 49kB | 33M | 1.91 |

has minimal storage costs while preserving most of the retrieval accuracy. Interestingly, we find out that quantizing the compressed tokens before storing them, then de-quantizing upon inference yields minimal performance degradation and can further improve storage-performance tradeoff. We refer the reader to Appendix H for more details.

## 5.3 ABLATION STUDIES

We conduct a series of ablation experiments to assess the robustness of *EDJE*, isolating the contributions of different design choices.

We begin by analyzing how varying the number of compressed tokens affects retrieval performance. Specifically, we evaluate Flickr30k zero-shot image retrieval using $\{32, 64, 128, 256\}$ target tokens in the token-compression adapter (Figure 4). As expected, increasing the number of tokens yields better performance, with a clear gap between the heavily compressed 32-token variant and the uncompressed "local" variant (576 tokens). Notably, 64 tokens strike an attractive balance between efficiency and retrieval quality. We additionally compare against simpler token-reduction strategies, providing alternative baselines to the proposed adapter; see Appendix G for details.

Next, we examine the sensitivity of *EDJE* to the size of the re-ranked pool $k$. Larger pools increase the likelihood of including relevant candidates but also introduce more distractors. It is therefore

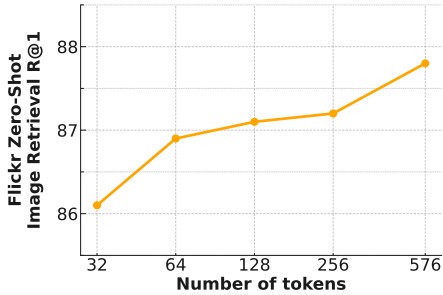

Figure 4: **Retrieval performance vs. number of tokens.** Flickr image retrieval for varying token counts, illustrating the compression–performance tradeoff.

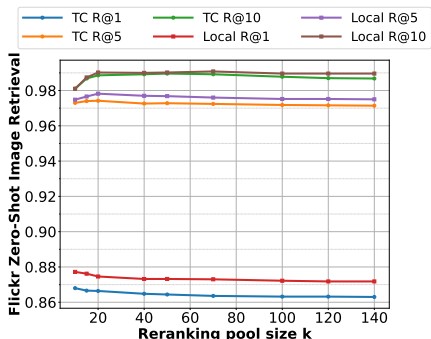

Figure 5: **Retrieval performance vs. reranking pool size.** Robustness of local and 64-token variants under different pool sizes on Flickr.

important to test the robustness of the re-ranker to different pool sizes and evaluate its tolerance to noise. We measure zero-shot retrieval under Recall@$\{1, 5, 10\}$ across varying pool sizes for both the local and 64-token variants (Figure 5). Results remain stable: while individual metrics fluctuate slightly, overall retrieval performance is consistent.

Finally, we ablate the pre-training objectives introduced in Section 4. For the local variant, we compare: (i) optimizing image–text matching (ITM) alone, (ii) ITM combined with masked language modeling (MLM), and (iii) the full objective that further adds text-embedding recovery. Each auxiliary loss contributes positively, with the full objective delivering the strongest results. We further evaluate the impact of local-to-compressed knowledge distillation, which provides further gains for compressed variants by transferring discriminative capacity from the local model. We also investigate cross-model negative selection to understand how *EDJE* behaves under different vision-encoder geometries. We refer the reader to Appendix B and Appendix E for more details.

## 5.4 Interpretable Semantics of Compressed Vision Tokens

To better understand the information preserved by the compressed visual tokens produced by *EDJE*, we aim to analyze their semantic structure regardless of any caption that may be equipped with the original image. We achieve this by inspecting the projection of each visual token into the language-model embedding space and assigning its nearest textual token from the LM vocabulary. Collecting these nearest neighbors (one per compressed tokens) reveals which language tokens the model most frequently associates with visual tokens, as depicted in Figure 6.

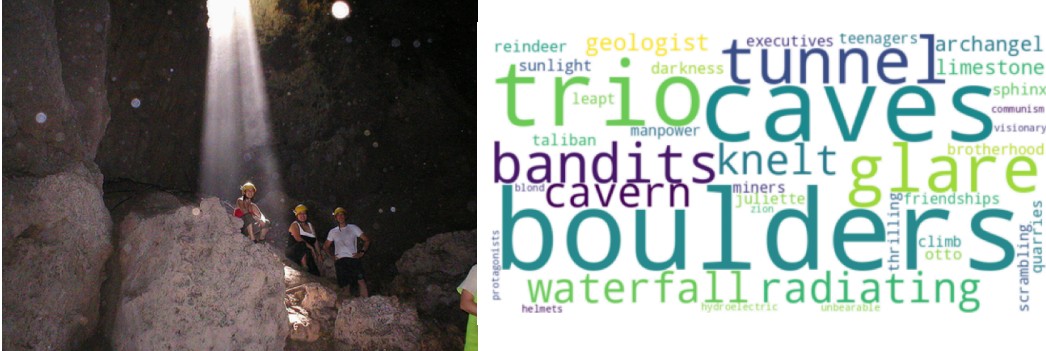

Figure 6: Semantic structure of the 64-token compressed representation. Left: example image from the Flickr-30k test set. Right: nearest-neighbor textual tokens assigned to compressed vision tokens; word size reflects frequency in the token distribution.

The compressed tokens map to meaningful object and scene descriptors such as *boulders*, *caves*, *glare*, or *trio*, indicating that the adapter preserves important semantic information. In contrast, we find that interestingly many of the uncompressed 576-token representation map to a meaningless special tokens (`unused80`), suggesting that a large portion of native ViT tokens carry redundant content for retrieval. We refer the reader to Appendix D for further explanations and experiments.

## 6 Discussion

We studied how to make joint vision–language re-rankers practical at scale. We recognize visual feature extraction as the key bottleneck in existing joint encoders. We tackle this problem by introducing *EDJE*. The approach of *EDJE* is to precompute the vision tokens and compress them with a lightweight adapter in an offline manner, in addition to a compact joint encoder that can deliver high-throughput inference while retaining high performance.

**Limitations and future work.** We think of this paper as a proof of concept that may inspire follow-up work; for instance, we did not cover multilingual-multimodal retrieval, which has drawn attention recently (Thapliyal et al., 2022) or other modalities such as audio or video. More broadly, we believe joint encoders are largely underexplored; putting effort into improving them can benefit a variety of applications including zero-shot classification and filtering large paired datasets.

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

# A  TRAINING HYPERPARAMETERS

Table A summarizes the main hyperparameters used throughout pre-training and fine-tuning. We fix the language model to MiniLM-L12-H384-uncased and freeze the vision encoder in all runs. Unless otherwise stated, all experiments use in-batch negative mining with three negatives per sample, and a re-ranking pool size of $k = 10$. These settings were chosen to balance training efficiency and retrieval quality and remained consistent across all backbones.

| Setting | Value |
| --- | --- |
| Language model | MiniLM-L12-H384-uncased |
| Adapter hidden dim | 8192 |
| Re-ranking pool size $k$ | 10 |
| Negatives per sample | 3 |
| Negative mining | In-batch, softmax-weighted top-$k$ |
| Hard negatives | None |
| Distillation | Sigmoid-BCE on pos/neg logits |
| Masking target | Caption tokens only |
| MLM masking probability | 0.5 |
| Mask excludes | Special tokens, image tokens |
| Truncation policy | `only_first` |
| Text max length | 64 |
| Batch size (evaluation) | 32 |
| Optimizer | AdamW |
| Weight decay | 0.05 |
| Pre-train LR | 3e-4 |
| Fine-tune LR | 2e-5 |
| Warmup steps | 100 |
| Warmup LR | 1e-6 |
| Min LR | 1e-6 |
| LR decay rate | 0.9 |
| Input resolution | 384 |

Table 3: Key hyperparameters, masking strategy, and negative sampling settings used in our experiments.

# B    ABLATION STUDY: TRAINING OBJECTIVES

We provide additional ablations on the training objectives. Table 4 shows the incremental effect of adding masked language modeling (MLM) and text–image contrastive learning (ITC) on top of the ITM baseline. Each component contributes positively to retrieval accuracy on Flickr30k with the SigLIP2-Large $384^2$ backbone, with ITM+MLM+ITC yielding the strongest results.

Table 5 focuses on token compression and highlights the effect of applying distillation. Here, knowledge distillation provides additional improvements when compressing visual tokens. This section analyzes the interpretability properties of the compressed visual tokens produced by EDJE. Although the token compression module discards a large portion of the original vision encoder sequence, we find that the resulting compact representation retains coherent semantic structure. We study two aspects: (1) the emergence of high-level concepts within individual compressed tokens, and (2) the alignment between compressed visual tokens and text features used during retrieval.

Table 4: **Ablation on training objectives.** We evaluate the effect of adding MLM and ITC on top of ITM. All configurations are evaluated on **SigLIP2 Large 384** backbone. Results are reported in terms of R@1 on Flickr30k.

| ITM | MLM | ITC | R@1 |
|:---:|:---:|:---:|:---:|
| ✓ | ✗ | ✗ | 82.3 |
| ✓ | ✓ | ✗ | 85.5 |
| ✓ | ✓ | ✓ | 87.8 |

Table 5: **Effect of distillation with token compression.** We report R@1 on Flickr30k using **SigLIP2 Large 384** with 64 tokens compression.

| Distillation | R@1 |
|:---:|:---:|
| ✗ | 83.8 |
| ✓ | 86.9 |

## C  USING *EDJE* IN PRACTICE

We provide a pseudo-code for using *EDJE* in practical retrieval pipelines, combining a clip-like embedding model, a vector store and *EDJE* as an efficient re-ranking model. A pseudo-code for the indexing stage is given in Algorithm 1 while the retrieval stage is summarized in Algorithm 2.

---

**Algorithm 1** Indexing (offline)

---

**Input:** Images (`images`), vector store (`store`), CLIP-like vision encoder (`vision_encoder`), finetuned *EDJE* token adapter (`adapter`)

```
 1: image_loader = DataLoader(images)
 2: for image_batch in image_loader do
 3:     // Extract both the usual embeddings and patch features in one pass
 4:     features, embeddings = vision_encoder(image_batch)
 5:     // Apply the token adapter on the encoders' features
 6:     features = adapter(features)
 7:     // Store embeddings on RAM and features on disk
 8:     store.insert(
 9:      embeddings=embeddings,
10:       extra_data={"features":  features}
11:     )
12: end for
```

---

**Algorithm 2** Retrieval

---

**Input:** Query text (`query`), vector store (`store`), CLIP-like text encoder (`text_encoder`), finetuned *EDJE* re-ranker (`re-ranker`), re-ranking pool size (`k`)

**Output:**  re-ranked retrieval results

```
 1: // Compute the usual query text-embedding
 2: text_embedding = text_encoder(query)
 3: // Retrieve candidates from vector store
 4: candidates = store.knn_query(text_embedding, k=k)
 5: // Compute re-ranking scores with EDJE
 6: features = torch.cat([candidate["features"] for candidate in
    candidates], dim=0)
 7: scores = re-ranker(query, features)
 8: results = candidates[scores.argsort(descending=True)]            return
    results
```

---

### C.1  TOKEN-FETCH I/O CONSIDERATIONS

Beyond compute, a practical deployment of *EDJE* must also account for the cost of fetching precomputed vision tokens from storage. In our experiments, each image is represented by 64 compressed tokens in BF16, corresponding to roughly $49\,\mathrm{kB}$ per image. For a re-ranking pool of 50k candidates, this amounts to reading approximately $2.46\,\mathrm{GB}$ of data per query.

To estimate the expected I/O overhead, we benchmark two storage layouts on a PCIe 4.0 NVMe SSD: (i) a single contiguous NumPy array storing all image representations, and (ii) a more realistic memory-mapped array accessed via random indices. In the contiguous case, loading the full $2.46\,\mathrm{GB}$ block takes around $0.39 \pm 0.003\,\mathrm{s}$ over 10 runs (approximately $6.3\,\mathrm{GB/s}$), which is in line with the advertised bandwidth of the SSD. With random access over a 100k-image memory-mapped index (50k random entries), the same amount of data is loaded in $0.59 \pm 0.04\,\mathrm{s}$.

These measurements indicate that, on modern local SSDs, the I/O cost of fetching compressed tokens is on the same order as the compute cost of the joint encoder and still allows end-to-end processing of tens of thousands of pairs per second. Note however that networked storage can exhibit substantially lower throughput, so we do not recommend it for *EDJE* deployments.

# D VISUAL TOKENS INTERPRETABILITY

## D.1 EMERGING VISION-TOKEN SEMANTICS

We next study the semantic content of the visual tokens produced by *EDJE* in a caption-independent manner. The goal is to qualitatively understand what information the compressed visual tokens carry, regardless of any ground-truth caption that may or may-not be provided.

Given an image, let $\{v_i\}_{i=1}^n$ denote either the compressed visual tokens (e.g, 64 tokens from the token-compression adapter) or the full ViT sequence (e,g 576 tokens, transformed locally). We first project each $v_i$ into the language-model embedding space using the same projection as in the joint encoder. For every visual token $v_i$, we then retrieve its nearest language token $w_i$ from the LM vocabulary (using cosine similarity). Collecting these nearest neighbors $\{w_i\}_{i=1}^n$ over the Flickr30k test set allows us to analyze which language-tokens the model most frequently associates with visual tokens.

This analysis has two desirable properties: (i) it evaluates the semantics of visual tokens purely through the geometry of the joint embedding space, without using the paired captions for that image, and (ii) it yields interpretable text tokens that can be visualized as through frequency histograms. We perform this analysis for two concrete example images for the compressed tokens, as illustrated in Figure 7.

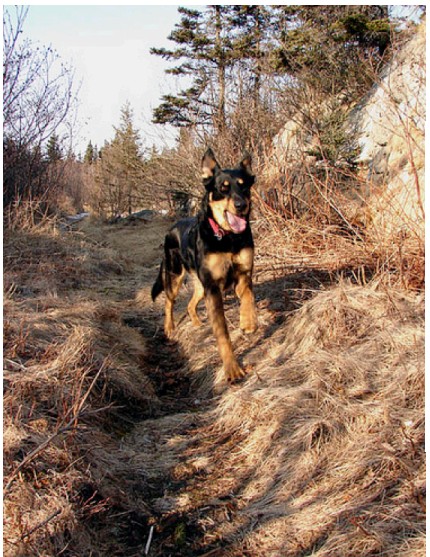 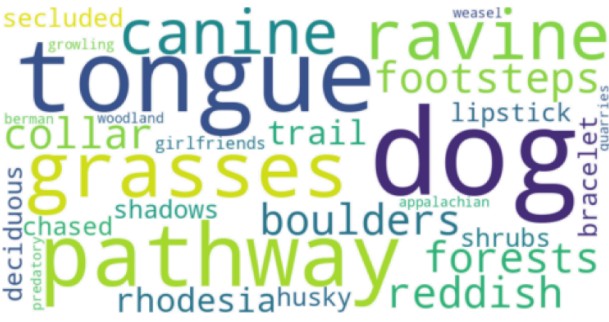

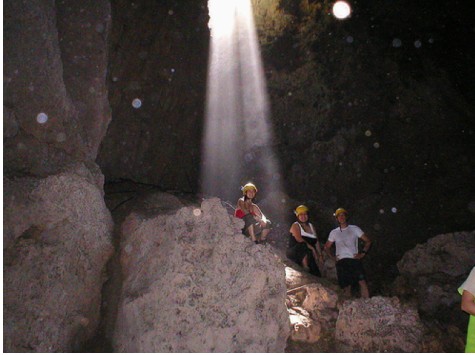 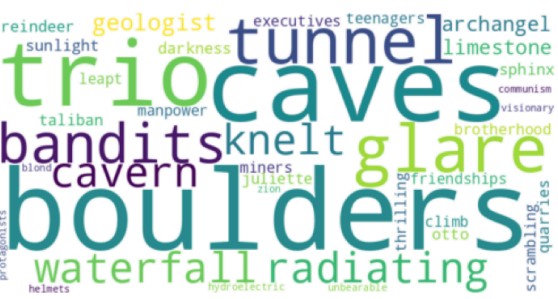

Figure 7: Emerging token semantics for the 64-token compressed representation. Left (per row): example images from Flickr test set. Right (per row): textual tokens word cloud, size indicates frequency according to vision token nearest-neighbors histogram.

The compressed tokens exhibit stable and highly interpretable semantics. Tokens frequently correspond to concrete object categories (e.g., dog, collar, boulders, trio) and scene attributes (e.g., glare, shadows). Despite the large reduction in sequence length, the model preserves a rich vocabulary of visual cues. We than repeated the analysis using the "local" *EDJE* variant, as illustrated in Figure 8.

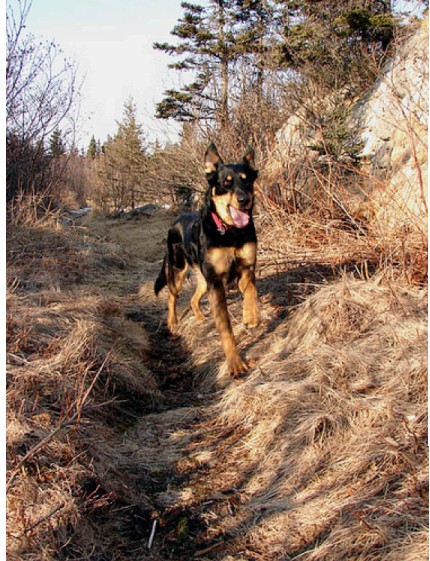
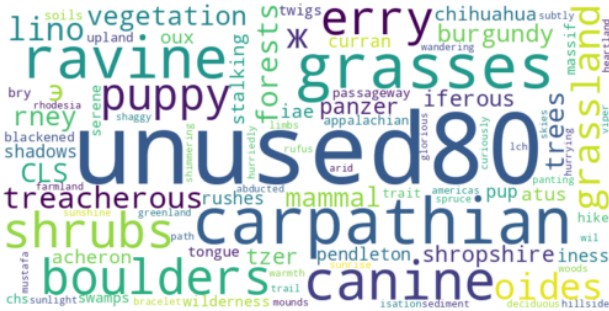

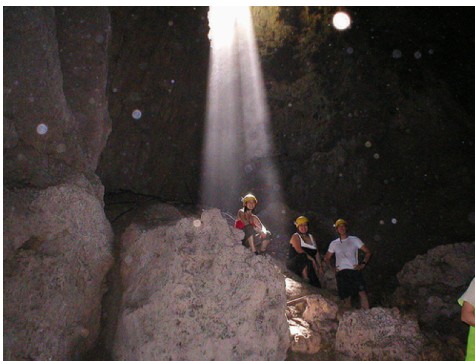
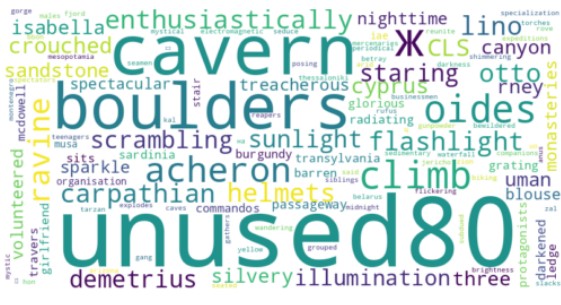

Figure 8: Token semantics for the 576-local *EDJE* representation. Left (per row): example images from Flickr test set. Right (per row): textual tokens word cloud, size indicates frequency according to vision token nearest-neighbors histogram.

In this case, however, a large fraction of tokens map to a special vocabulary item, unused80, or to scattered low-frequency words, resulting in much less concentrated distributions. This suggests that many of the original ViT tokens have low semantic content in the joint space and are largely redundant for downstream retrieval.

### D.2 Image-Caption Token Alignment

We complement the caption-independent analysis above with a quantitative study of how well the compressed visual tokens align with caption tokens. The goal is to measure to what extent cross-modal alignment is preserved when reducing the number of visual tokens from the full ViT sequence to the compressed *EDJE* representation.

For each image-caption pair, we first preprocess the caption by lowercasing all words and removing punctuation and standard English stopwords (e.g., "A child playing in the ocean." → "child playing ocean").

Let $\{t_j\}_{j=1}^m$ denote the remaining textual tokens and $\{v_i\}_{i=1}^n$ the set of visual tokens for the corresponding image, where $\{v_i\}$ is either the full (locally transformed) ViT sequence (e.g., $n = 576$) or the compressed set produced by *EDJE* (e.g., $n = 64$).

We embed both text tokens and visual tokens into the joint embedding space using the same projections as in the joint encoder, and compute a ColBERT-like alignment score (Khattab and Zaharia, 2020; Faysse et al., 2025). For each textual token $t_j$, we compute its maximum cosine similarity to any visual token,

$$s_j = \max_{1 \leq i \leq n} \cos\big(f(t_j), g(v_i)\big),$$

where $f(\cdot)$ and $g(\cdot)$ are the corresponding text and vision projections. We then define the *alignment score* for the image–caption pair as the average of these maxima,

$$S = \frac{1}{m} \sum_{j=1}^m s_j.$$

Intuitively, $S$ measures how well each caption word can "find" at least one strongly related visual token in the image. Note that we didn't directly optimize $S$ as a scoring criterion.

We compute the distribution of alignment scores $S$ over all Flickr30k test pairs for both the full 576-token representation and the 64-token compressed representation as depicted in Figure 9.

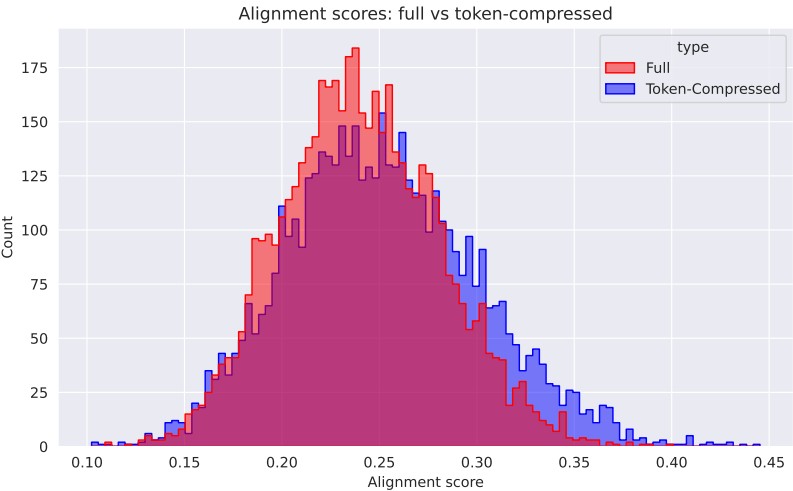

Figure 9: Distribution of alignment scores for full token sequences and their token compressed counterparts. The histogram shows that compression preserves the overall alignment structure while slightly shifting the distribution toward lower but still tightly concentrated scores. This supports the claim that the compressed representations retain most of the semantic signal needed for retrieval.

The token-compressed representation achieves an average alignment score of $0.2516 \pm 0.0491$, while the full-sequence representation achieves $0.2404 \pm 0.0405$. The corresponding histograms overlap and exhibit very similar shapes, with the compressed representation showing a slightly higher mean. A two-sided paired t-test on the per-pair scores indicates that this difference is statistically significant ($p = 3.0 \times 10^{-153}$), with a small-to-moderate effect size ($d = 0.39$). Overall, these results show that

*EDJE* 's 64-token representation preserves, and slightly improves, image-caption alignment relative to the full 576-token ViT sequence, despite using $9\times$ fewer visual tokens.

# E    CROSS MODEL NEGATIVE MINING

In the main experiments, hard negatives for *EDJE* are mined using the same embedding model family as the underlying retriever (e.g., SigLIP2 for a SigLIP2-based *EDJE*). To assess the impact of this design choice, we additionally study *cross-model* negative mining, where the miner and the backbone come from different embedding models.

Concretely, we consider two settings: (i) using SigLIP2 as a negative miner for a CLIP-based *EDJE*, and (ii) using CLIP as a negative miner for a SigLIP2-based *EDJE*. In both cases, the joint encoder and token-compression components remain fixed; only the model used to select hard negatives during fine-tuning is changed.

Table 6: Using SigLIP2 as a negative miner for a CLIP based EDJE. Mining with SigLIP2 does not substantially affect CLIP performance.

| Model | Flickr Zero-Shot | | COCO Finetuned | |
|---|---|---|---|---|
| | Text-to-Image | Image-to-Text | Text-to-Image | Image-to-Text |
| CLIP ViT L/14@336 | 81.9 | 93.8 | 54.6 | 70.7 |
| +SigLIP2 L miner | 81.8 | 93.5 | 54.6 | 70.8 |

Table 7: Using CLIP as a negative miner for a SigLIP2 based EDJE. Mining with CLIP significantly degrades SigLIP2 performance.

| Model | Flickr Zero-Shot | | COCO Finetuned | |
|---|---|---|---|---|
| | Text-to-Image | Image-to-Text | Text-to-Image | Image-to-Text |
| SigLIP2 ViT L/16@384 | 87.8 | 96.5 | 64.9 | 81.0 |
| +CLIP L miner | 81.5 | 93.0 | 58.9 | 74.48 |

The results in Tables 6 and 7 reveal an interesting pattern. When SigLIP2 is used as a negative miner for a CLIP-based *EDJE*, performance changes only marginally: Flickr30k zero-shot R@1 remains essentially unchanged, and COCO fine-tuned scores are within a similar range. This suggests that in this setting, overall performance is largely limited by the vision backbone rather than by the precise geometry of the negative miner. In contrast, when CLIP is used as a miner for a SigLIP2-based EDJE, performance drops substantially on both Flickr30k and COCO, for both text-to-image and image-to-text retrieval. This degradation emphasizes the importance of selecting sufficiently hard negatives. Overall, these experiments support the practical guideline that hard-negative mining should be performed with a model that is at least as strong as, and well aligned with, the underlying vision-encoder used in *EDJE*.

## F    FULL DATASET RETRIEVAL EVALUATION

To provide a more comprehensive evaluation, we adopt the full-dataset retrieval protocol used in LightningDOT (Sun et al., 2021), where retrieval is performed against all images and captions in the dataset, including the train/validation splits. This setting is considerably more challenging and better reflects real-world retrieval scenarios. We follow LightningDOT's setup for both Flickr Full and COCO Full, and scale the re-ranking pool size to 100 in all experiments.

Table 8: Flickr Full retrieval results under the LightningDOT setup. Retrieval is performed against all dataset's images/captions as retrieved instances. *EDJE* is evaluated in a zero-shot setting and substantially outperforms LightningDOT in both text-to-image and image-to-text retrieval.

| Model | Text-to-Image | | | Image-to-Text | | |
|---|---|---|---|---|---|---|
| | R@5 | R@10 | R@20 | R@5 | R@10 | R@20 |
| LightningDOT | 60.1 | 69.5 | 78.3 | 75.1 | 83.9 | 90.5 |
| *EDJE* | 78.32 | 84.54 | 89.58 | 92.4 | 95.9 | 97.7 |

Table 9: COCO Full retrieval results under the LightningDOT setup. Retrieval is performed against all dataset's images/captions as retrieved instances. *EDJE* significantly outperforms LightningDOT across all recall levels and directions.

| Model | Text-to-Image | | | Image-to-Text | | |
|---|---|---|---|---|---|---|
| | R@5 | R@10 | R@20 | R@5 | R@10 | R@20 |
| LightningDOT | 37.3 | 46.8 | 56.4 | 48.0 | 59.0 | 68.9 |
| *EDJE* | 52.23 | 60.55 | 68.08 | 69.86 | 76.96 | 82.64 |

As shown in Tables 8 and 9, *EDJE* substantially improves full-dataset retrieval performance over LightningDOT on both Flickr Full and COCO Full. On Flickr Full (zero-shot), *EDJE* yields large gains in both directions and at all recall levels (e.g., text-to-image R@5 improves from 60.1 to 78.32, and image-to-text R@5 from 75.1 to 92.40). On COCO Full (fine-tuned), *EDJE* again surpasses LightningDOT by a wide margin (e.g., text-to-image R@5 from 37.3 to 52.23, image-to-text R@5 from 48.0 to 69.86). These results confirm that *EDJE* remains effective in more realistic large-candidate retrieval scenarios.

## G    TOKEN COMPRESSION BASELINES

In this section we compare the proposed token-compression adapter in *EDJE* to several alternative ways of reducing the number of visual tokens produced by the SigLIP2 ViT-L encoder. All methods start from the same 576-token ViT sequence (384×384 resolution) and compress it to 64 tokens per image. We evaluate on Flickr30k (zero-shot) and COCO (fine-tuned).

We consider the following token-compression strategies:

1. **Striding.** A simple subsampling baseline that keeps every 9th token from the 576-token ViT sequence, yielding 64 tokens in total ($576/9 = 64$).

2. **Token clustering.** We run k-means++ over the 576 visual tokens for each image to obtain 64 clusters, and use the cluster centroids as compressed tokens.

3. **Attention pruning.** We compute the attention scores of each token in the last ViT layer and keep the 64 most attended tokens (where attendance is averaged across heads).

For fairness, each configuration is pretrained and fine-tuned end-to-end with the same protocol as our token-compression model. Particularly, we distill from the uncompressed (576-token) teacher as in the main *EDJE* experiments.

Table 10: Comparison of token-compression strategies applied to the SigLIP2 ViT-L image encoder. We report Recall@1 on Flickr30k (zero-shot) and COCO (fine-tuned). *SigLIP2 (Baseline)* denotes the original embedding-only model. All other methods compress the 576 ViT tokens to 64 tokens per image and use the same joint encoder architecture.

| Model | Flickr Zero-Shot | | COCO Finetuned | |
|---|---|---|---|---|
| | Text-to-Image | Image-to-Text | Text-to-Image | Image-to-Text |
| SigLIP2 (Baseline) | 82.3 | 94.8 | – | – |
| Striding | 83.24 | 94.1 | 60.09 | 77.2 |
| Clustering | 85.66 | 96.1 | 63.31 | 79.76 |
| Attention Pruning | 82.4 | 93.7 | 58.6 | 76.4 |
| *EDJE* | **86.9** | **96.4** | **64.6** | **80.9** |

We observe that *EDJE*'s token-compression adapter consistently outperforms striding, clustering, and attention pruning across both datasets and directions (text-to-image and image-to-text). This emphasizes the superiority of the learned query-based adapter over generic pooling or pruning schemes.

# H    QUANTIZING THE VISION TOKENS

In all main experiments, the precomputed vision tokens are stored in BF16 (or FP16) format. Because *EDJE* is designed for large-scale retrieval with potentially billions of stored image representations, it is interesting to examine how far the storage footprint can be reduced via quantizing the numerical precision of the vision tokens.

We perform post-training quantization only on the precomputed vision tokens, while keeping the joint encoder itself in BF16. For each quantization type, the cached tokens are quantized on disk and de-quantized immediately before being fed to the joint encoder at inference time, with no additional finetuning.

We consider two representation families: (i) the full 576-token ViT sequence, and (ii) the 64-token compressed representation. For each family we evaluate three numeric formats: (a) BF16 (no quantization), (b) FP8-E4M3, and (c) FP4-E2M1. We measure Flickr30k zero-shot image retrieval performance (R@1) and compute the average storage per image in kilobytes, taking into account the number of tokens, hidden dimension, and numeric precision. This allows experimentation with a wide range of formats including FP4 and 1-bit signed vectors.

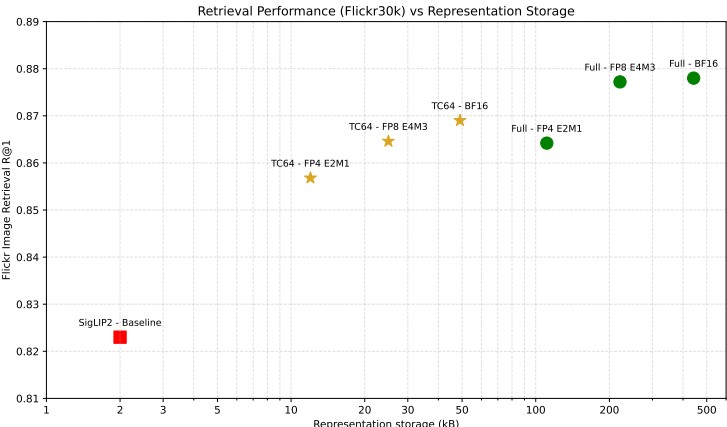

Figure 10: Retrieval performance (Flickr30k zero-shot image retrieval Recall@1) versus image representation storage size (in kilo-bytes, log scaled). We compare the full 576-token ViT representation and the 64-token compressed representation under BF16, FP8-E4M3, and FP4-E2M1 formats. *EDJE* 's compressed tokens retain strong performance even under aggressive quantization, while substantially reducing storage.

Figure 10 summarizes the trade-off between retrieval performance and representation storage. The plot shows that the token-compression is remarkably robust to aggressive quantization: moving from BF16 to FP8 and even FP4 leads to only minor changes in R@1, while substantially reducing storage. Moreover, combining FP8 quantization with 64-token compression cuts the per-image storage in half beyond the already compact 49 kB, with negligible loss in retrieval quality, pushing the storage-performance tradeoff even further.

# I ON EXTENDING *EDJE* TO VIDEO RETRIEVAL

Although video retrieval lies outside the scope of this work, *EDJE* naturally admits several extensions to the video domain. A video may be represented as a sequence of frames $[f_1, \ldots, f_T]$, each processed independently by the vision encoder to produce $\boldsymbol{X}_t = g_\phi(f_t) \in \mathbb{R}^{n \times d}$. Concatenating per-frame features yields a high-dimensional tensor

$$\mathbf{X} = [\boldsymbol{X}_1, \ldots, \boldsymbol{X}_T] \in \mathbb{R}^{T \times n \times d},$$

corresponding to $T \times n$ visual tokens. The central challenge is therefore to compress hundreds of frames efficiently while preserving temporal structure and producing a compact representation suitable for re-ranking.

Below, we outline two concrete strategies for adapting *EDJE* to video-text retrieval.

1. **_EDJE_'s adapter as a temporal compressor.** A direct extension is to repurpose the *EDJE* token-compression adapter to operate *across time*. Each frame is first mapped to a single visual token using any image embedding model, yielding a temporal sequence of $T$ tokens. A temporal adapter can then be trained to compress these tokens into a small set of temporally aggregated tokens. This design captures long-range semantics across many frames and improves upon simple temporal pooling strategies such as averaging frame embeddings (Rasheed et al., 2022; Maaz et al., 2023). Temporal positional encodings can be incorporated by adding a learned position vector to each frame token before passing it to the adapter.

2. **Spatial compression followed by temporal compression.** An improvement over the aforementioned idea is to enhance single frame representations by replacing naive embeddings by compressing frames via a pretrained version of *EDJE*'s token-compression adapter, applied to each frame to better compress information spatially. Then, a similar token-compression adapter is learned to temporally aggregate all of the compressed tokens from each frame in an efficient manner (e,g $576 \rightarrow 64$ per frame).

## J    IMPACT OF TEXT-ENCODER CHOICE ON RETRIEVAL

The joint encoder in *EDJE* uses an extremely lightweight MiniLM-L12-H384 text model for joint encoding. A natural question is whether replacing MiniLM with a larger, more expressive language encoder such as BERT-Base could further improve retrieval performance. Prior work in vision language modeling suggests that stronger language encoders sometimes offer marginal gains in caption understanding, but they also introduce greater computational cost and may provide diminishing returns in retrieval settings where the visual features dominate (Devlin et al., 2019; Radford et al., 2021; Jia et al., 2021; Li et al., 2022).

To study this trade-off, we replace MiniLM with *BERT-Base* (uncased) while keeping all other components fixed. In particular, we use the same SigLIP2 Base visual encoder, the same token-compression module, and the same training protocol used in our main experiments. The joint encoder therefore differs only in the text backbone. We evaluate this configuration on Flickr30k using the standard zero-shot retrieval protocol.

Table 11: Flickr30k zero-shot retrieval results when substituting MiniLM with BERT-Base as the text encoder in the joint module. The visual encoder is SigLIP2-Base in all configurations. *EDJE* (MiniLM) denotes the baseline from the main paper.

| Text Encoder | Text-to-Image | | | Image-to-Text | | |
|---|---|---|---|---|---|---|
| | R@1 | R@5 | R@10 | R@1 | R@5 | R@10 |
| MiniLM with Siglip Base | 84.3 | **96.6** | 97.9 | **94.3** | 99.9 | 99.9 |
| BERT-Base with Siglip Base | **84.52** | 96.58 | 97.9 | 93.9 | 99.9 | 99.9 |

As shown in Table 11, substituting MiniLM with BERT-Base produces mixed but notable effects. BERT-Base yields slight improvements in the text-to-image setting, particularly on R@1 and R@10 (e.g., R@1 increases from $84.3$ to $84.52$, and R@10 from $97.9$ to $97.98$). In contrast, MiniLM remains marginally stronger for image-to-text retrieval (e.g., R@1 improves from $93.9$ with BERT to $94.3$ with MiniLM), while both models tie at higher recall levels.

Although these results do not indicate that larger language encoders provide substantial gains in our setup, it is important to note that our model is trained on only 12M images, considered small-scale relative to typical vision-language pretraining regimes. Nevertheless, MiniLM, despite its compact size, does not appear to bottleneck the token-compressed retrieval pipeline.

Overall, this experiment shows that BERT-Base can slightly improve specific aspects of retrieval, but lightweight, well-distilled text encoders like MiniLM remain highly competitive and more efficient for *EDJE*'s joint-encoder design.

