# OpenReview forum: "Efficient Discriminative Joint Encoders for Large Scale Vision-Language Reranking"
_ICLR.cc/2026/Conference — ICLR 2026 Poster_

### Official Review · Reviewer_EbYo · 2025-10-27

**Soundness:** 3
**Presentation:** 3
**Contribution:** 2
**Rating:** 6
**Confidence:** 4

**Summary:**

The paper studies the problem of visual-language retrieval. More specifically, the paper focuses on the re-ranking stage of visual-language retrieval. To this end, the authors propose a new architecture, i.e., efficient discriminative joint encoder, to do the re-ranking efficiently. Besides, the authors propose a lightweight token-compression adapter that condenses vision features to reduce storage and computation. Experiments show the re-ranking method can consistently achieve a gain for the text-image retrieval performance. Analysis demonstrates the efficiency of the proposed method.

**Strengths:**

1. I like the idea of the paper. The design of the architecture enhances the efficiency of re-ranking in a smart way.

2. Performance is good with efficiency. The experiments show the proposed method achieves a consistent performance gain in re-ranking, while being highly efficient.

3. Clear presentation of the paper. The paper is clearly presented. The teaser figure is good, together with several highlighted 'key takeaways', making the important ideas of the paper very clear.

**Weaknesses:**

1. Scope of the paper. Although the title of the paper is 'vision-language', the paper only studies text-image retrieval, and the video-text retrieval, which is a very important and more challenging task, has not been touched. It would be better if the paper can also include the results for video-text retrieval so that it is more general and will have a wider audience for the ICLR conference.

2. Small issues: 'rerank' should be 're-rank' through out the entire paper.

**Questions:**

Could the method also be applied to video-text retrieval and have the same conclusion as it is for text-image retrieval? This is very important concerning the scope and significance of the paper.

---

> ### Author Response · Authors · 2025-11-22
> **Response to reviewer EbYo**
>
> Thank you very much for your review. We truly appreciate the positive feedback on our method, including your comments regarding the novelty of the architecture design, the strong efficiency–performance trade-off, and the clarity of the paper’s presentation. We are grateful that you highlighted these strengths, and they have motivated us to further refine and expand the work in the revised version.
>
> Regarding the minor issue you noted (“rerank” -> “re-rank”), thank you for pointing this out. We have corrected this throughout the first revised version of the manuscript.
>
> Concerning your request for video–text retrieval experiments: this is an excellent suggestion, and we fully agree that video-text retrieval is an important and impactful direction. During the rebuttal period, we have been working hard running a large number of additional experiments - including full-dataset retrieval benchmarks, expanded token-compression baselines, and model-capacity analyses - which are summarized in the new Appendices C-H.
>
> That said, we absolutely recognize the value of evaluating our method in the video domain. We are currently working toward increasing the broader applicability and impact of EDJE, including multilingual retrieval and generalization to video-text retrieval. We will do our best to include at least a minimal proof-of-concept experiment for video-text retrieval before the camera-ready version.
> We appreciate your review and are happy to answer any further questions.

---

> ### Author Response · Authors · 2025-11-29
> **Update regarding applicability of EDJE to video**
>
> While it was infeasible to conduct the text-to-video retrieval experiments within the limited rebuttal period, in particular given the substantial effort required to perform the additional experiments and analyses requested, we believe that EDJE can be naturally extended to the video domain, possibly without even modifying the architecture. To illustrate this, we include two concrete extension strategies in **Appendix I** of the revised paper. We believe that this addition sheds light on the potential of EDJE to be extended to other multimodal tasks, highlighting its flexibility, its relevance for real-world video understanding scenarios, and its broader impact on the design of general multimodal retrieval systems.

---

### Official Review · Reviewer_bquk · 2025-10-29

**Soundness:** 3
**Presentation:** 3
**Contribution:** 3
**Rating:** 6
**Confidence:** 4

**Summary:**

The paper introduces EDJE, an efficient discriminative joint encoder for large‑scale vision–language reranking. EDJE precomputes vision tokens offline, compresses them via a lightweight attention‑based adapter into a small set of tokens, and feeds these with text into a compact language model at query time. On Flickr30k (zero‑shot) and COCO (fine‑tuned), EDJE matches or approaches prior joint encoders while drastically reducing online latency and storage (e.g., 1.9-4.1 ms per 64-sample batch; 49 kB/image with 64 tokens). The paper includes ablations on token count, pool size, and training objectives.

**Strengths:**

- Moves ViT compute offline; integrates vision via a compact joint encoder, addressing the main blocker for joint reranking.

- Universal‑query adapter reduces hundreds of ViT tokens to tens while preserving accuracy; 64 tokens ($\sim$49 kB/image) is a strong trade‑off.

- EDJE consistently improves Flickr30k ZS R@1, e.g., CLIP ViT‑L/14@336: 67.7->81.9 (T2I).

- With SigLIP2‑L/16@384, EDJE‑Local gets 87.8/96.5 (Flickr T2I/I2T R@1) at 4.14 ms (64‑batch), vs BLIP‑2's 88.6/96.9 at ~99 ms.

- Objectives (ITM+MLM+ITC) and distillation help; pool‑size sensitivity is stable.

**Weaknesses:**

- Latency excludes disk I/O/feature fetch; the 50k pairs/s figure from the abstract lacks a pipeline breakdown.

- Only Flickr30k ZS and COCO FT; no tests under domain shift, web‑scale noise, or multilingual queries despite claims of modularity.

- EDJE is competitive but not strictly superior to BLIP‑2 on R@1; the paper could more precisely state where it leads/trails.

- Lacks comparisons to alternative compression/pooling (e.g., Perceiver‑style latents, strided token dropping) to justify the chosen adapter beyond results shown.

- Storage numbers imply FP16 tokens; quantization (FP8/INT8) and its impact aren’t analyzed.

**Questions:**

- Can you report end‑to‑end throughput/latency (including token fetch from SSD/NVMe, k‑reranking, and aggregation) to contextualize the 50k pairs/s claim?

- How sensitive is performance to storage format/hardware (e.g., memory‑mapped arrays vs individual files; SATA vs NVMe vs network storage)? Any prefetching?

- What is the effect of quantizing compressed tokens (FP8/INT8) on R@k and latency? Can training be made quantization‑aware?

- Negative mining currently uses the same embedding model family; how does EDJE fare with cross‑model negatives to reduce bias toward the retriever geometry?

- Beyond varying token count, can you vary adapter capacity (depth/width) to separate capacity vs token‑count effects at 32–64 tokens?

- Any preliminary multilingual results (e.g., Crossmodal-3600) to support future-work claims?

---

> ### Author Response · Authors · 2025-11-22
> **Response to reviewer bquk (part 1/2)**
>
> We thank the reviewer for the thoughtful and constructive feedback on our work. We have uploaded a first revised version of the manuscript that already incorporates substantial new experiments and analyses motivated by your comments and those of the other reviewers, including new results on end-to-end throughput and storage, token-compression baselines, quantization of compressed tokens, and cross-model negative mining (see new Appendices C, E, G, and H). In what follows, we address your points in detail. In parallel, we are running additional experiments (e.g., adapter-capacity ablations and preliminary multilingual results), and we will upload a second revised version including these further results before the end of the rebuttal period.
> Thank you for recognizing the empirical significance of our work. Your feedback helped us broaden the experiments and motivated additional analyses on latency, storage, and quantization.
>
> ***
>
> **Throughput, I/O, and hardware sensitivity.**
> The 50k pairs/s figure reported in the abstract refers to pure joint encoding throughput, i.e., assuming vision tokens are already in the GPU’s memory. This isolates the throughput efficiency of EDJE relative to prior joint encoders. While disk I/O drastically varies with the hardware setup, we agree that adding context on I/O is indeed helpful. We have run small additional experiments on our machine to give a sense of the end-to-end picture.
>
> On our PCIe4 SSD system, we measure I/O for 64-token compressed representations (49 kB/image, 2.46 GB for 50k images):
>
> 1. **Contiguous block, single .npy file (best-case sequential read):** We simulate an “ideal” layout where 50k image representations are stored contiguously in a single NumPy array. Loading this block takes 0.39 ± 0.0033 [sec] (10 runs) for a total size of 2.46GB.  This corresponds to a speed of ~6.3 GB/sec which is reasonable for our setup.
>
> 2. **Random access over 100k-image index using `np.memmap`:** We simulate a 100k-image index stored as a memory-mapped .npy file and sample 50k random indices to fetch. In this setting, loading 2.46GB of compressed representations took 0.59 ± 0.0438 [sec] (10 runs).
> These numbers suggest that, on a standard modern SSD, I/O is of comparable magnitude to the re-ranking compute and still allows end-to-end processing of tens of thousands of pairs per second. In contrast, performance for networked storage can vary widely (from hundreds of MB/s to multiple GB/s depending on the configuration), so we view SSD-backed storage as the most realistic and recommended deployment scenario for EDJE. We refer the reviewer to appendix C in the first revised version of the paper for further details.
>
> ***
>
> **Token compression baselines.**
> We thank the reviewer for the comment that was also made by R1.
> We have added the following baselines, compressing the SigLIP2 ViT-L output from 576 tokens to 64 tokens in all cases:
> 1.	Simple striding: keep every 9-th token (576/9 = 64).
> 2.	Token clustering: k-means++ over the 576 tokens into 64 clusters; centroids are used as compressed tokens.
> 3.	Attention pruning: keep the 64 tokens with highest average attention score in the last ViT layer.
> For fairness, each configuration is pretrained and fine-tuned end-to-end with the same protocol as our token-compression model, and we distill from the uncompressed (576-token) teacher as in the original EDJE experiments.
> We report Recall@1 on Flickr (zero-shot) and COCO (fine-tuned):
>
> | Model | Flickr Zero-Shot | | COCO finetuned |  |
> | --- | --- | --- | --- | --- |
> | | Text-to-Image | Image-to-Text | Text-to-Image | Image-to-Text |
> | SigLIP2 (Baseline) | 82.3 | 94.8 | -- | -- |
> | Striding | 83.24 | 94.1 | 60.09 | 77.2 |
> | Clustering | 85.66 | 96.1 | 63.31 | 79.76 |
> | Attention Pruning | 82.4 | 93.7 | 58.6 | 76.4 |
> | **EDJE** | **86.9** | **96.4** | **64.6** | **80.9** |
>
> Across both datasets, EDJE’s token compression method consistently outperforms pooling, clustering, and attention-pruning alternatives, confirming that the learned query-based adapter is not only storage-efficient but also empirically stronger.
> We refer the reviewer to **Appendix G** in the first revised version of the paper for further details.

---

> ### Author Response · Authors · 2025-11-22
> **Response to reviewer bquk (part 2/2)**
>
> **Quantizing the tokens.**
> Thank you for this valuable suggestion. We ran additional experiments where we quantize only the compressed vision tokens, leaving the language model in FP16. For each quantization scheme, we quantize the cached tokens and de-quantize them just before feeding them to the joint encoder. This setup isolates the effect of quantizing the stored representation and allows us to explore precisions beyond FP8, such as FP4.
>
> Concretely, we experimented with FP8-E4M3 and FP4-E2M1 formats and evaluated Flickr30k zero-shot image retrieval (R@1) as a function of representation size for both “full” and token-compressed variants; we refer the reviewer to **Appendix H** of the first-revised paper for the full results.
>
> We make two primary observations: (i) performance is remarkably robust to aggressive quantization - even with FP4 quantization, R@1 drops only minimally compared to the FP16 baseline, without any quantization-aware training; and (ii) the storage-performance tradeoff improves further, as combining FP8 quantization with 64-token compression cuts the per-image storage in half beyond the already compact 49 kB, with negligible loss in retrieval quality.
> Latency is essentially unchanged in our setup, since de-quantization is a lightweight operation compared to running the joint encoder.
>
> ***
>
> **Cross-model negative mining.**
> We appreciate this question and ran additional experiments with cross-model negative miners.
> Specifically, we consider two settings during fine-tuning: SigLIP2 miner for a CLIP-based EDJE, and a CLIP miner for a SigLIP2-based EDJE.
> The results (Flickr30k zero-shot and COCO fine-tuned) are:
> | Model                 | Flickr Zero-Shot T2I | Flickr Zero-Shot I2T | COCO Finetuned T2I | COCO Finetuned I2T |
> |-----------------------|----------------------|----------------------|--------------------|--------------------|
> | CLIP ViT L/14@336     | 81.9                 | 93.8                 | 54.6                  | 70.7                  |
> | +SigLIP2 L miner      | 81.8                 | 93.5                 | 54.6               | 70.8               |
>
>
> | Model                    | Flickr Zero-Shot T2I | Flickr Zero-Shot I2T | COCO Finetuned T2I | COCO Finetuned I2T |
> |--------------------------|----------------------|----------------------|--------------------|--------------------|
> | SigLIP2 ViT L/16@384     | 87.8                 | 96.5                 | 64.9               | 81.0               |
> | +CLIP L miner            | 81.5                 | 93.0                 | 58.9               | 74.48              |
>
>
> We observe that (i) using SigLIP2 to mine negatives for a CLIP-based EDJE has very little impact on performance, suggesting that in this regime the overall performance is largely bounded by the weaker CLIP backbone; and using CLIP as a miner for SigLIP2-based EDJE significantly degrades performance, indicating that a strong miner that matches (or exceeds) the underlying backbone is important. In this case, CLIP’s geometry produces negatives that are suboptimal for training a SigLIP2-based re-ranker.
>
> We refer the reviewer to **Appendix E** in the first revised version of the paper for further details.
> Adapter capacity vs. token count (32–64 tokens)
> We agree that disentangling adapter capacity from token count is important for understanding the design space. We are currently running experiments that vary the width and depth of the MLP used in the adapter. We expect to complete them within the rebuttal period. We will notify the reviewer and will include the results in the revised version as soon as possible.
>
> ***
>
> **Preliminary multilingual results.**
> We will work on these promptly and notify you if we manage to have them within the rebuttal period.
>
> ***
>
> All of the additional experiments and analyses described above (I/O and end-to-end throughput measurements, token-compression baselines, quantization of compressed tokens, and cross-model negative mining) are already integrated into the currently uploaded revised manuscript, in particular in Appendices C, E, G, and H. During the rebuttal period, we have worked hard to significantly broaden the empirical evaluation and to better characterize EDJE’s behavior under realistic deployment constraints (storage, I/O, quantization) in direct response to your suggestions. We are continuing to run the remaining experiments on adapter capacity (for different token counts) and, if time permits, preliminary multilingual evaluations, and will include these in a further updated version before the rebuttal deadline.
> We hope that these additions address your concerns about the practical relevance, robustness, and design choices of our method. If you feel that the revised manuscript and the newly added experiments satisfactorily resolve your points, we would be very grateful if you could consider reflecting this in your overall assessment.

---

> > ### Author Response · Authors · 2025-11-29
> > **Update regarding experiments**
> >
> > We have completed an additional experiment concerning **model capacity versus accuracy (larger language models)** related to **capacity vs token count**. The results have been added to **Appendix J** in the revised version of the paper.

---

### Official Review · Reviewer_tFqx · 2025-10-30

**Soundness:** 3
**Presentation:** 3
**Contribution:** 2
**Rating:** 4
**Confidence:** 3

**Summary:**

The authors propose an Efficient Discriminative Joint Encoder (EDJE) to address the deployment challenges of vision-language reranking models in large-scale scenarios due to computational constraints. Multimodal retrieval typically incurs higher inference costs than text retrieval due to visual computation, making efficient interaction between textual and visual information a key factor in determining the practicality of multimodal retrieval models. The authors alleviate the aforementioned issue to some extent by further compressing visual information into a more compact token sequence, which proves effective according to the experimental results. However, the novelty of this work and the interpretability of the model require further clarification.

**Strengths:**

The research focuses on the demand for efficient retrieval in the current multimodal retrieval field, which has strong practical significance. It proposes a vision-language reranking model that addresses the deployment challenges of multimodal retrieval joint encoders and provides new insights for multimodal retrieval methods. Experimentally, EDJE demonstrates competitive performance while significantly reducing storage requirements and online computational costs, achieving high-throughput inference. The core concepts of the paper are clearly defined, with no ambiguous expressions or logical contradictions.

**Weaknesses:**

The proposed framework essentially follows the well-established two-stage retrieval paradigm, where visual features are pre-extracted offline and a lightweight joint model is used for re-ranking. While this design improves efficiency, it does not introduce a fundamentally new retrieval or representation mechanism. Similar strategies have been adopted in prior works such as LightningDOT [1] and VISTA [2]. The token-compression adapter in EDJE, as described, functions primarily to further compress visual encoder outputs into a compact set of tokens for offline storage and later consumption by the language model; this operational choice—offline compression of visual features into a small set of learned tokens—is conceptually similar to prior approaches such as TokenLearner [3] and the Q-Former used in BLIP-2 [4]. Consequently, the methodological novelty appears limited: EDJE mainly integrates established components (offline encoding, attention-based token compression, and a lightweight re-ranker) for an engineering trade-off between storage and online compute.

[1] Sun, Siqi, et al. "Lightningdot: Pre-training visual-semantic embeddings for real-time image-text retrieval." Proceedings of the 2021 Conference of the North American Chapter of the Association for Computational Linguistics: Human Language Technologies. 2021.

[2] Zhou, Junjie, et al. "Vista: Visualized text embedding for universal multi-modal retrieval." arXiv preprint arXiv:2406.04292 (2024).

[3]Ryoo, Michael, et al. "Tokenlearner: Adaptive space-time tokenization for videos." Advances in neural information processing systems 34 (2021): 12786-12797.

[4] Li, Junnan, et al. "Blip-2: Bootstrapping language-image pre-training with frozen image encoders and large language models." International conference on machine learning. PMLR, 2023.

**Questions:**

1. How does the token-compression adapter preserve or improve multimodal alignment between the compressed visual tokens and textual representations?

2. If a different vision encoder is used, must the adapter be retrained and all previously stored tokens regenerated? Could the authors comment on the practical implications of this design choice?

3. How is the semantic quality of the compressed visual features assessed? Would visualizations or qualitative analysis of the compressed tokens help to evaluate information retention?

4. After compression by the adapter, how interpretable are the visual tokens? Are there methods to analyze or explain what visual information each token represents?

---

> ### Author Response · Authors · 2025-11-22
> **Response to reviewer tFqx (part 1/2)**
>
> We thank the reviewer for the detailed and insightful comments on our work. We have uploaded a first revised version of the manuscript that already incorporates substantial new experiments and clarifications requested by you and the other reviewers, including extended analyses of semantic interpretability, multimodal alignment (see new Appendices C–H in the paper). In what follows, we address your comments point by point. In parallel, we are running additional experiments, and we will upload a second revised version including these further results before the end of the rebuttal period.
>
> ***
>
> **Relation to LightningDOT.**
> While we were originally not aware of LightningDOT, we would like to clarify the conceptual differences between LightningDOT and EDJE, as well as the additional contribution of EDJE beyond reusing the general idea of re-ranking.
> In the LightningDOT paper, the authors utilized models such as UNITER and OSCAR to perform re-ranking. In these models, images are represented by a single embedding vector for each R-CNN region, and only a handful of such region vectors are used (e.g., K=5). While this approach is what made storing the representations possible, it severely limits the expressiveness of the representation. Importantly, because each region is collapsed into a single vector, such re-ranker behaves much closer to an embedding model rather than a true joint encoder that sees the full image. In practice, this leads to performance that now lags behind modern embedding models such as SigLIP2.
>
> In contrast, papers such as ALBEF (published in the same year) show that using the full image token sequence enables significantly stronger retrieval performance, although at infeasible storage costs. EDJE directly tackles this trade-off: it leverages full ViT representations but introduces a compact, data-driven token compression mechanism that preserves expressiveness while drastically reducing storage requirements. As a result, EDJE achieves retrieval accuracy comparable to full-sequence joint encoders while remaining efficient and scalable. We observe this both quantitatively (performance on COCO and Flickr) and qualitatively (see interpretability analysis).
> We agree that LightningDOT is an important prior work and will include it in the related work section of the revised paper, clarifying the above distinctions.
>
> **Relation to VISTA, TokenLearner, and Q-Former.** VISTA primarily focuses on building an embedding model for universal/composed retrieval (multimodal queries and targets), producing a single embedding per (multimodal) input.  TokenLearner, in turn, performs token reduction inside the vision backbone to cut FLOPs for image/video recognition; the selected tokens are recomputed online for every input. While our proposed token-compression method draws some inspiration from Q-Former, BLIP-2 employs Q-Former layers as a cross-attention layers after every self-attention layer, significantly increasing the parameters and computations during the joint encoding stage.
>
> ***
>
> **Interpreting the semantic quality of the compressed tokens.**
> We thank the reviewer for this insightful comment, which led us to several meaningful observations regarding the semantic interpretability of the compressed visual features.
> To assess the semantic quality of the tokens, we first consider the projection of each compressed visual token into the language-model embedding space. For every visual token, we then retrieve its nearest language token from the LM vocabulary and analyze these nearest neighbors over the Flickr test set using histograms and word-cloud visualizations. An advantage of this technique is that it allows to interpret the semantic quality of the tokens independently of any paired caption.
> We refer the reviewer to **Appendix D.1** in the first-revised version of the paper for concrete examples and further details.
> We find that the compressed tokens consistently correspond to clear semantic concepts, whereas many of the original 576 ViT tokens map to a special “unused80” token, indicating low semantic content. This suggests that the adapter successfully removes redundant or low-information tokens while concentrating meaningful visual information into a compact set.

---

> > ### Author Response · Authors · 2025-11-22
> > **Response to reviewer tFqx (part 2/2)**
> >
> > **Multimodal alignment after compression.**
> > To quantify alignment, we evaluate how well the compressed visual tokens preserve correspondence with caption tokens on the Flickr test set.
> > We first preprocess captions by lowercasing and removing punctuation and stopwords.
> > Then, for each textual token - we compute its maximum cosine similarity to any visual token from the associated image. Averaging these maxima yields an alignment score for each image–caption pair. We compute this score both using the full set of 576 ViT tokens and using the 64 compressed tokens produced by EDJE, allowing us to directly compare how much cross-modal alignment is retained after compression.
> >
> > We obtain an average alignment score of 0.2404 ± 0.0405 for the full-sequence representation and 0.2516 ± 0.0491 for the compressed representation. The difference is statistically significant (according to a two-sided paired t-test) with a small-to-moderate effect size (Cohen’s d=0.39). We refer the reviewer to **Appendix D.2** for further details and analyses. Overall, these results show that EDJE ’s 64-token representation preserves, and slightly improves, image-caption alignment relative to the full 576-token ViT sequence, despite using 9× fewer visual tokens.
> >
> > We believe this result supports the view that the adapter manages to remove non-semantic tokens and compress those that matter for cross-modal alignment.
> >
> > ***
> >
> > **Practical implications of changing the vision encoder.**
> > Yes, if the vision encoder is changed, the token-compression adapter must be retrained and the compressed tokens regenerated. Because the adapter learns to compress the specific token distribution produced by a given vision encoder, switching to a different encoder requires retraining the adapter, and all previously stored compressed tokens must be regenerated. This is entirely expected: regenerating representations when replacing the underlying model is standard practice in retrieval systems. For example, upgrading an embedding model (e.g., replacing CLIP with SigLIP2) similarly requires recomputing all stored embeddings. Regarding the retraining cost, one advantage of EDJE is that - due to its high efficiency and small parameter count - the adapter is extremely cheap to train.
> >
> > ***
> >
> > All of the above analyses (including the new semantic-interpretability and alignment studies) are integrated into the currently uploaded revised manuscript, in particular in the new Appendices C–H. During the rebuttal period, we have worked hard to significantly extend the experimental section and to better position EDJE with respect to prior work, aiming to directly address your concerns about novelty, semantic quality of the compressed tokens, cross-modal alignment, and practical implications. We will upload a further updated version with updated prior work and more results before the rebuttal deadline. We are willing to address any remaining issues.
> >
> > If you feel that the added experiments and clarifications resolve your concerns, we would be very grateful if you could consider reflecting this in your overall assessment.

---

> ### Comment · Reviewer_tFqx · 2025-11-24
>
> I thank the authors for their detailed response. My concerns have been addressed, and the experimental results have alleviated my previous doubts. Therefore, I recommend accepting this paper. Additionally, I suggest making the code publicly available, which would greatly benefit the community and enhance the paper's impact.

---

> > ### Author Response · Authors · 2025-11-24
> > **Response to reviewer tFqx**
> >
> > Dear reviewer tFqx, we are pleased to see that we have properly addressed your concerns!
> > We will put the link to our (anonymous) repository in the next revised version.

---

> > > ### Author Response · Authors · 2025-12-01
> > >
> > > Dear Reviewer tFqx,
> > >
> > > Following your request, the final revised version of the manuscript now includes a link to our anonymous repository.
> > >
> > > Thank you again for your helpful suggestion.
> > >
> > > Best regards,
> > >
> > > The Authors

---

### Official Review · Reviewer_kbcQ · 2025-10-31

**Soundness:** 2
**Presentation:** 3
**Contribution:** 2
**Rating:** 4
**Confidence:** 2

**Summary:**

The paper introduces EDJE, a system that makes image–text search faster and more accurate by running a small joint model on precomputed and compressed image features. Unlike existing models that are too slow to use in large-scale systems, EDJE speeds up inference by more than 50× while keeping similar accuracy. It improves search results across various models and datasets without requiring extra fine-tuning.

**Strengths:**

- The paper introduces a practical method EDJE for image-text reranking by combining precomputed vision features with an cross-attention-based token compression adapter. The token compression design is well-motivated.
- Empirical validation are conducted across multiple vision backbones and datasets. The proposed method achieves competitive or superior retrieval accuracy with drastically improved efficiency.

**Weaknesses:**

Major weakness:
- Although the performance is strong, the retrieval benchmarks are quite outdated. To mimc closer to real-world retrieval setting, I would suggest the authors to follow

    [1] Sun, Siqi, et al. "Lightningdot: Pre-training visual-semantic embeddings for real-time image-text retrieval." Proceedings of the 2021 Conference of the North American Chapter of the Association for Computational Linguistics: Human Language Technologies. 2021.

    This paper introduces retrieval across the full COCO/FLICKR dataset to better mimc real-world retrieval setting.

    [2] Jiang, Ziyan, et al. "Vlm2vec: Training vision-language models for massive multimodal embedding tasks." arXiv preprint arXiv:2410.05160 (2024).

    This paper introduces a more realistic benchmark MMEB where the retrieval query and target are both multimodal.

- In addition, the reranking system conceptually is similar to Lightningdot, although the implementation details are quite different and now inspired by latest advanced in VLMs.

- EDJE drastically improves in zero-shot retrieval. However, on the fine-tuned COCO retrieval task, EDJE is only on par with prior models. One could argue that a larger-capacity joint encoder with more parameters might still have an advantage in the fine-tuned regime.The accuracy vs. model size trade-off is not fully explored in this paper. It would be helpful if the paper explore using a larger language model or more parameters in EDJE.

Minor weakness:
- All citations should use \citep{} instead of \cite{}
- The term “local variant” vs. “token-compressed variant” is used to distinguish the full-token vs. compressed EDJE, but the term “local” isn’t very self-explanatory. If I understand correctly, it means the MLP adapter outputs the full local tokens without compression.

**Questions:**

- As I pointed above, I would suggest the authors to expand the experiments to more realistic evaluation settings.
- For token compression, what about the following baselines?
    - Simple pooling (e.g., mean pooling)
    - Direct token pruning techniques (like token clustering, attention pruning) on the vision tokens, similar to methods used in ViT efficiency papers, as an alternative to query-based compression.
    - What if we adopt the pre-trained Q-former from BLIP and keep it frozen?

---

> ### Author Response · Authors · 2025-11-22
> **Response to reviewer kbcQ (part 1/2)**
>
> We thank the reviewer for the thorough and constructive feedback on our work. We have uploaded a first revised version of the manuscript that already incorporates substantial new experiments and clarifications suggested by you and the other reviewers (see the new Appendices C–H). Below we summarize the main changes and respond to your comments in detail. In parallel, we are running additional experiments (e.g., with a larger language model), and we will upload a second revised version including these results before the end of the rebuttal period.
>
> ***
>
> **Evaluation Benchmarks**
> We agree that evaluating on the full COCO / Flickr30k test sets, as in LightningDOT, is a more realistic setting than the commonly used 1k-image splits. Following your suggestion, we added experiments on Flickr-Full and COCO-Full, using LightningDOT’s evaluation protocol and scaling the reranking pool size to 100 for both datasets.
>
> Flickr-Full results:
> | Model | | Text-to-Image | | | Image-to-Text | |
> | --- | --- | --- | --- | --- | --- | --- |
> |  | R@5 | R@10 | R@20 | R@5 | R@10 | R@20 |
> | LighningDOT | 60.1 | 69.5 | 78.3 | 75.1 | 83.9 | 90.5 |
> | EDJE | 78.32 | 84.54 | 89.58 | 92.4 | 95.9 | 97.7 |
>
>
> COCO-Full results:
> | Model | | Text-to-Image | |  | Image-to-Text |  |
> | --- | --- | --- | --- | --- | --- | --- |
> |  | R@5 | R@10 | R@20 | R@5 | R@10 | R@20 |
> | LighningDOT | 37.3 | 46.8 | 56.4 | 48.0 | 59.0 | 68.9 |
> | EDJE | 52.23 | 60.55 | 68.08 | 69.86 | 76.96 | 82.64 |
>
> We refer the reviewer to appendix F in the first-revised version of the paper for further details.
>
> **MMEB.**
> MMEB is primarily designed for the composed retrieval task, where either query or target consist of both modalities. Our work focuses on standard image-text retrieval, which is the regime EDJE is specifically designed and optimized for. We therefore view MMEB as complementary and outside the present scope. We will add these new full-dataset results and a discussion of MMEB to the revised version.
>
> ***
>
> **Relation to LightningDOT**
> While we were originally not aware of LightningDOT, we would like to clarify the conceptual differences between LightningDOT and EDJE, as well as the additional contribution of EDJE beyond reusing the general idea of re-ranking.
>
> In the LightningDOT paper, the authors utilized models such as UNITER and OSCAR to perform re-ranking. In these models, images are represented by a single embedding vector for each R-CNN region, and only a handful of such region vectors are used (e.g., K=5). While this approach is what made storing the representations possible, it severely limits the expressiveness of the representation. Importantly, because each region is collapsed into a single vector, such re-ranker behaves much closer to an embedding model rather than a true joint encoder that sees the full image. In practice, this leads to performance that now lags behind modern embedding models such as SigLIP2.
>
> In contrast, papers such as ALBEF (published in the same year) show that using the full image token sequence enables significantly stronger retrieval performance, although at infeasible storage costs. EDJE directly tackles this trade-off: it leverages full ViT representations but introduces a compact, **data-driven token compression** mechanism that preserves expressiveness while drastically reducing storage requirements. As a result, EDJE achieves retrieval accuracy comparable to full-sequence joint encoders while remaining efficient and scalable. We observe this both quantitatively (performance on COCO and Flickr) and qualitatively (we kindly refer you to interpretability analysis in our response to reviewer tFqx).
> We agree that LightningDOT is an important prior work and will include it in the related work section of the revised paper, clarifying the above distinctions.
>
> ***
>
> **Model capacity vs. accuracy (larger language models)**
> We agree that this is an important question and are actively running experiments with larger language models to better understand the size–performance trade-off. In particular, we are currently conducting experiments with a BERT-base encoder to match the language-model used in prior work (ALBEF, BLIP, BLIP-2). These runs are still in progress, and we expect them to complete within the rebuttal period; we will update the reviewers with the corresponding results and include them in the revised version.

---

> ### Author Response · Authors · 2025-11-22
> **Response to reviewer kbcQ (part 2/2)**
>
> **Token compression baselines**
> We thank the reviewer for the excellent suggestions on token-compression baselines.
> We have added the following baselines, compressing the SigLIP2 ViT-L output from 576 tokens to 64 tokens in all cases:
> Simple striding: keep every 9-th token (576/9 = 64).
> Token clustering: k-means++ over the 576 tokens into 64 clusters; centroids are used as compressed tokens.
> Attention pruning: keep the 64 tokens with highest average attention score in the last ViT layer.
> For fairness, each configuration is pretrained and fine-tuned end-to-end with the same protocol as our token-compression model, and we distill from the uncompressed (576-token) teacher as in the original EDJE experiments.
> We report Recall@1 on Flickr (zero-shot) and COCO (fine-tuned):
> | Model | Flickr Zero-Shot | | COCO finetuned |  |
> | --- | --- | --- | --- | --- |
> | Model | Text-to-Image | Image-to-Text | Text-to-Image | Image-to-Text |
> | SigLIP2 (Baseline) | 82.3 | 94.8 | -- | -- |
> | Striding | 83.24 | 94.1 | 60.09 | 77.2 |
> | Clustering | 85.66 | 96.1 | 63.31 | 79.76 |
> | Attention Pruning | 82.4 | 93.7 | 58.6 | 76.4 |
> | **EDJE** | **86.9** | **96.4** | **64.6** | **80.9** |
>
>
> Across both datasets, EDJE’s token compression method consistently outperforms pooling, clustering, and attention-pruning alternatives, confirming that the learned query-based adapter is not only storage-efficient but also empirically stronger.
> We refer the reviewer to appendix G in the first revised version of the paper for further details.
>
> **On using a frozen BLIP-2 Q-Former.**
> We also considered the suggestion of reusing a pre-trained Q-Former from BLIP-2 as our token-compression module, but this is unfortunately not straightforward for two main reasons. First, the BLIP-2 Q-Former is trained jointly with a different vision encoder and language model than our SigLIP2 + MiniLM setup, so plugging it in “as is” would require additional projection layers and likely substantial retraining to resolve the mismatch in feature spaces. Second, in BLIP-2 the Q-Former is implemented as cross-attention layers interleaved with self-attention within a multimodal transformer. It is therefore unclear which layer should be used as a standalone compression module.
>
> ***
>
> All of the above additional experiments and analyses will be incorporated into the revised version of the paper. We hope these additions and clarifications address your concerns regarding benchmark realism, novelty, model capacity, and token-compression design and are willing to address any other questions. If you feel they have satisfactorily addressed your concerns, we would kindly ask you to consider reflecting this in your overall score.

---

> > ### Author Response · Authors · 2025-11-29
> > **Update regarding required experiments**
> >
> > We have completed the final experiment requested by the reviewer concerning **model capacity versus accuracy (larger language models)**. The results have been added to **Appendix J** in the revised version of the paper.

---

### Author Response · Authors · 2025-11-29
**General response**

Dear AC and Reviewers,
We thank all four reviewers for their thorough and constructive evaluations. We greatly appreciate the time invested in engaging with our work, and we have substantially revised the paper in response to their feedback, including extensive new experiments, analyses, and clarifications. We were pleased to see that Reviewer tFqx raised their score and voted for acceptance, which indicates that we have resolved all of their concerns. The other three reviewers didn't manage to respond before the reviewer-discussion phase was suddenly ended.

Most of the rebuttal-driven changes appear in Appendices C–J and are marked in $\textcolor{blue}{\text{blue}}$ in the revised version of the paper.
Below we summarize the main concernes and questions raised by the reviewers, indicate who raised each point, and briefly describe how they are addressed in the revised paper.
We refer the AC to the reviewers’ comments for more detailed responses.


## Experiments conducted

1. **Evaluation benchmarks.** we extended the breadth of the evaluation benchmarks by evaluating on the Flickr/COCO-Full benchmarks, included in  **Appendix F** in the revised paper (as requested by reviewer kbcQ).
2. **Model capacity vs. accuracy (larger language models).** we conducted an ablation study with a larger-language model (bert-base), and included its results in **Appendix J** of the revised paper (as requested by reviewer kbcQ and bquk).
3. **Token compression baselines.** We added three additional baselines for token-reduction methods, and included the corresponding results in **Appendix G** (as requested by reviewers kbcQ and bquk).
4. **Interptation and Alignment of the compressed tokens.** We added both qualitative and quantitative interpretation and alignment analyses in **Appendix D.1** and **Appendix D.2** (raised by reviewer tFqx, who acknowledged our response on this point, recommending acceptance).
5. **Throughput, I/O, and hardware sensitivity.** We contextualize the running time of EDJE by explicitly including I/O times through two distinct experiments, reported in **Appendix C** of the revised paper (raised by reviewer bquk).
6. **Quantizing the compressed tokens.** We experiment with different quantizations of the compressed tokens (FP8-E4M3 and FP4-E2M1), evaluating retrieval performance when quantizing the tokens and de-quantizing them at inference time. The results are included in **Appendix H** of the the revised paper, as requested by reviewer bquk.
7. **Cross-model negative mining.** We explored negative mining with a different vision model and report two experiments in **Appendix E** in the revised version of the paper.


## Comparison to other papers
8. **Relation to LightningDOT and other papers.** we elaborated on the relation to the LightningDOT paper in the "related work" section of the revised paper. Concerns about novelty were raised by reviewers kbcQ and tFqx; our response resolved this concern for reviewer tFqx, who replied before the review period closed. We also clarified our relation to other relevant works (VISTA, TokenLearner, and Q-Former), which was acknowledged by reviewer tFqx.

## Extension to other modalities
9. **Extension to text-to-video retrieval.** we included two distinct approaches to natuarlly extend EDJE to text-to-video retrieval in **Appendix I** of the revised paper (raised by reviewer EbYo)

## Minor fixes
10. **Minor formatting issues.** we fixed  'rerank -> re-rank' throughout the entire paper (raised by reviewer EbYo) and citation style; \cite{} -> \citep{} (raised by  reviewer kbcQ).

---

### Meta-Review · Area_Chair_i1gP · 2026-01-03

**Summary:**

The submission initially received mixed reviews (4466). The main concerns can be summarized into the following points:
1. The retrieval benchmarks are quite outdated; more comprehensive evaluations on extra benchmakrs are needed.
2. The key idea is not new; it is similar to several prior works (such as LightningDOT or VISTA).
3. More ablations are expected. For example, the trade-off between accuracy and model size.
4. Extra explanations about the semantic quality of the compressed visual features, and compressed visual tokens are necessary.

In the rebuttal, all those concerns have been well-addressed, and thus I recommend Accept.

**Reviewer Concerns:**

All concerns are well-addressed in the rebuttal.

**Reviewer Scores:**

I think all two "negative" reviewers will increase their ratings to 6, and the left two reviewers will maintain their inital positive ratings.

---

### Decision · Program_Chairs · 2026-01-26

Accept (Poster)